

# REDCAPP (v1.0): Parameterizing valley inversions in air temperature data downscaled from re-analyses

Bin Cao[1,2], Stephan Gruber[2], and Tingjun Zhang[1]

[1]Key Laboratory of Western China's Environmental Systems (MOE), College of Earth and Environmental Sciences, Lanzhou University, Lanzhou 730000, China
[2]Department of Geography & Environmental Studies, Carleton University, Ottawa, K1S 5B6, Canada

*Correspondence to:* Bin Cao (caob08@lzu.edu.cn)

**Abstract.** In mountain areas, the use of coarse-grid re-analysis data for driving fine-scale models requires downscaling of near-surface (e.g. 2 m high) air temperature. Existing approaches describe lapse rates well but differ in how they include surface effects, i.e. the difference between the simulated 2 m and upper-air temperatures. We show that different treatment of surface effects result in some methods making better predictions in valleys while others are better in summit areas. We propose the downscaling method REDCAPP (REanalysis Downscaling Cold Air Pooling Parameterization) with a spatially variable magnitude of surface effects. Results are evaluated with observations (395 stations) from two mountain regions and compared with three reference methods. Our findings suggest that the difference between near-surface air temperature and pressure-level temperature ($\Delta T$) is a good proxy of surface effects. It can be used with a spatially-variable Land-Surface Correction-Factor ($LSCF$) for improving downscaling results, especially in valleys with strong surface effects and cold air pooling during winter. While $LSCF$ can be parameterized from a fine-scale digital elevation model (DEM), the transfer of model parameters between mountain ranges needs further investigation.

## 1 Introduction

Air temperature ($T$) controls a variety of environmental processes (Jones and Kelly, 1983). Predicting $T$ at fine-scale, however, is challenging in hilly and mountainous terrain because the lateral variability of $T$ is larger and subject to a greater diversity of processes than in gentle terrain. Direct observations of $T$ are usually sparse in mountains (Daly, 2006; Minder et al., 2010) and correspondingly, their interpolation is often not a reliable basis for estimating $T$ over larger areas. Atmospheric reanalyses, which are produced by assimilating observational data into numerical weather prediction model runs (Kistler et al., 2001; Dee et al., 2011; Harris et al., 2014), are a valuable alternative as their output is available on regular grids. In oder to make predictions at the fine scale ($\sim$10–100 m) required for representing topography, the coarse-scale ($\sim$10–100 km) reanalysis data need to be downscaled (Bürger et al., 2012). Previous studies (Fiddes and Gruber, 2014; Gupta and Tarboton, 2016; Gao et al., 2012) have reported how reanalysis data can be used to represent the elevation dependency of $T$ in downscaling. This study investigates how to further refine corresponding predictions and outlines a REanalysis Downscaling Cold Air Pooling Parameterization (REDCAPP).

Approaches for downscaling can be classified as: (1) dynamical, using physically-based models; and (2) statistical, using



empirical-statistical relationships (Bürger et al., 2012). Typically, regional climate models (RCMs) are used for dynamical downscaling aimed at deriving fine-scale data consistent with large-scale climate fields. As RCMs are computationally expensive, their spatial resolution is often restricted to ∼1–10 km (Hay and Clark, 2003; Maraun et al., 2010; Hagemann et al., 2004). Additionally, the lack of appropriate parameterizations or numerical methods often restricts how finely resolved RCMs can be run in mountains (Kiefer and Zhong, 2015). Statistical methods make fine-scale predictions based on statistical or empirical relationships between observations and coarse-scale fields (Yang et al., 2012). Statistical downscaling usually is computationally efficient (Chu et al., 2010; Hofer et al., 2010; Souvignet et al., 2010) but the requirement for observations inherent in many methods limits their applicability to mountains and remote areas.

A number of downscaling methods have been proposed that rely on physically-based empirical-statistical relationships and thus do not require local station data (Fiddes and Gruber, 2014; Gao et al., 2012). The basic assumption of these methods is that vertical gradients imposed by topography are more important than horizontal ones. The simplest method, here referred to as REF1 (reference method 1), uses a fixed lapse rate, usually $-6.5\,°C\,km^{-1}$ (Dimri, 2009; Giorgi et al., 2003), for describing the elevation dependence of $T$. Lapse rates are reported to be variable (Blandford et al., 2008; Lundquist and Cayan, 2007; Minder et al., 2010) and many of the drivers of this variability are represented in reanalysis models. Upper-air temperature, described at different pressure levels ($T_{pl}$) in reanalyses, has been used to derive average lapse rates over large areas through linear regression against geopotential or elevation (Mokhov and Akperov, 2006; Gruber, 2012). Recently, Fiddes and Gruber (2014) presented $T$ downscaling through direct interpolation of $T_{pl}$ (REF2), and Gao et al. (2012) obtained fine-scale $T$ by adding a lapse rate derived from $T_{pl}$ to surface air temperature ($T_{sa}$) (REF3).

While REF2 and REF3 have achieved some successes owing to the strong and well-described influence of elevation on $T$, they differ in their treatment of surface effects. The ground surface warms or cools near-surface air with respect to the upper-air temperature. For this reason, reanalyses provide separate variables for $T_{sa}$ (surface air temperature) and $T_{pl}$ (upper-air temperature at several pressure levels). Surface effects in mountains, however, are spatially heterogeneous. It is obvious that a peak, having only a small area of ground surfaces in proximity, will on average be subject to much weaker surface effects than a valley. Additionally, during periods of strong radiative cooling, the lateral drainage of cold air can lead to cold air pooling (CAP) in valley bottoms, further differentiating surface effects spatially. For example, Lewkowicz and Bonnaventure (2011) reported that average lapse rates could be positive in mountains due to strong winter inversion and result in lower $T$ in valleys than at higher locations. In the reference methods, surface effects on $T$ are either ignored (REF2) or treated as spatially invariant at the fine scale (REF1 and REF3). It is thus desirable to find a way to describe the spatial and temporal patterns of surface effects in mountainous terrain and to incorporate it into downscaling parameterization schemes.

In this study, we describe and test a method (REDCAPP) for parameterizing the temporal and spatial differentiation of surface effects and cold air pooling when downscaling reanalysis data in mountainous areas. The method is based on deriving a proxy of surface effects ($\Delta T$) from reanalysis data and then adding it, in spatially varying amounts, to the fine-scale air temperature derived from pressure levels. This is accomplished with a "land surface correction factor" ($LSCF$) estimated based on terrain morphometry. Specifically, we address four research questions: (1) Is $\Delta T$ suitable for parameterizing CAP and surface effects? (2) How well can we estimate $LSCF$ from a fine-scale digital elevation model (DEM)? (3) How much does REDCAPP





improve downscaling when compared with reference methods? (4) Can REDCAPP parameters easily be transferred between different mountain ranges? In this study, we describe REDCAPP and its application with ERA-Interim data. We investigate patterns of $\Delta T$ spatially and in time series using differing topographic locations, such as deep valleys, slopes and peaks. We then compare $LSCF$ fitted to station data with estimates derived from fine-scale DEMs. The performance and transferability

of REDCAPP are evaluated using a large number of observations from the Swiss Alps and the Chinese Qilian Mountains in the north-east of the Qinghai-Tibetan Plateau.

## 2   Background

### 2.1   Near-surface and upper-air temperature

In this study, the difference between near-surface air temperature $T_{sa}$ and upper-air temperature $T_{pl}$ is important. Upper-air

refers to the portion of atmosphere well above the Earth's surface, which is gently stirred towards the large-scale forcing field and in which the effects of the land surface friction on the air motion is negligible (Van De Berg and Medley, 2016). In reanalyses, upper-air variables are typically available at discrete vertical levels defined in terms of air pressure and ranging from near sea level to tens of kilometres height. This makes $T_{pl}$ a four-dimensional variable (longitude, latitude, pressure level, time) and it is given also at pressure levels corresponding to elevations lower than the model topography. The near-surface air

temperature $T_{sa}$ is directly influenced by the land surface via its energy balance and roughness. Reanalysis data is produced by coupled atmosphere-land-ocean models, which usually represent upper air temperature and land-surface conditions rather well (Compo et al., 2011). Since $T_{sa}$ and $T_{pl}$ are available in reanalysis products, the strength of the simulated land-surface effects on $T_{sa}$ can be quantified by their difference ($\Delta T$).

Fiddes and Gruber (2014) interpolate values from pressure levels to obtain the upper-air temperature at the elevation of the

fine-scale topography $T_{pl}^{f}$. Assuming that in each time step only the magnitude of land surface effects varies at the fine scale (Parsons and Daly, 1983), $\Delta T$ can be added back to $T_{pl}^{f}$ after multiplication with the land surface correction factor $LSCF$. The following Sections 2.2 and 2.3 describe the rationale of predicting $LSCF$ from DEM-derived geomorphometric variables.

### 2.2   Land surface effects and hypsometric position

With increasing altitude, the influence of local circulation is gradually transferred to regional circulation. By consequence, $T$ is

more strongly controlled by land-surface effects in low areas such as valley bottoms and almost exclusively by the upper air at high elevations such as mountain peaks (Tabony, 1985). This means hypsometric position can be used as a geomorphometric proxy for the relative strength of land-surface effects. The hypsometric position $[0, 1]$ refers to the cumulative density of fine-scale elevation being higher than a given location within a defined surrounding area.





## 2.3 Cold air pooling

CAPs, also known as "valley inversion" or "temperature inversion", occur in topographic depressions and often, the air near the surface is colder there than the air above (Lareau et al., 2013). CAP is caused by downslope flow and accumulation of cold air (Kiefer and Zhong, 2015), usually during periods of strong radiative cooling (Lareau et al., 2013). The temperature

inversion can vary from 1 °C to more than 10 °C depending on the surrounding terrain (e.g. land cover and valley geometry) and weather situation (Kiefer and Zhong, 2015; Whiteman et al., 2001). CAPs are common in almost all sizes of basins and valleys (Kiefer and Zhong, 2015; Mahrt et al., 2001), and their strength is expected to be related to how low and sheltered valleys are (Lareau et al., 2013). In order to predict CAPs at the fine scale based on $\Delta T$, a geomorphometric variable is needed for identifying valleys and for comparing the "degree of valleyness".

## 3  Data

### 3.1  ERA-Interim

ERA-Interim is a global reanalysis product produced by the European Center for Medium-range Weather Forecast (ECMWF) using a fully coupled atmosphere-ocean-land model and four-dimensional variational assimilation (Berrisford et al., 2011). It has 60 pressure levels in the vertical, with the top level at 1 mb. A reduced Gaussian grid with approximately uniform 79 km

spacing for surface and other grid-point fields is used. ERA-Interim data covers the period from 1 January 1979 onward and are extended with current observations with little delay (Dee et al., 2011). ERA-Interim produces four analyses per day at 00:00, 06:00, 12:00 and 18:00 UTC for the surface and 60 pressure levels in the upper atmosphere. ERA-Interim has been evaluated for various mountain regions via field measurements and proved to resolve large-scale climate well (Bao and Zhang, 2013; Mugford et al., 2012; Fiddes et al., 2015; Hodges et al., 2011; Chen et al., 2014). In this study, 2-meter temperature and air

temperatures of the lowermost sixteen pressure levels covering 1000–500 mb (with respect to an elevation range of ∼100–6000 m a.s.l) are used as $T^c_{sa}$ and $T^c_{pl}$ (see Appendix A for subscript/superscript conventions).

### 3.2  Observations and quality control

The observational mean daily air temperatures ($T_{obs}$) from the Swiss Alps and the Qilian Mountains are used for deriving model parameters and for evaluating results (Table 1, Figure 1). Observation datasets from the Swiss Alps were obtained

from the MeteoSwiss automatic monitoring network (184 stations) and from the Inter-cantonal Measurement and Information System (IMIS) at the WSL Institute for Snow and Avalanche Research SLF (178 stations). In the Qilian Mountains, there are 30 stations from the Heihe Watershed Allied Telemetry Experimental Research (HIWATER) and 3 stations from the Third Pole Environment Database (TPED) (Li et al., 2013). Temperatures are observed by automatic meteorological stations using intervals from 10 to 30 minutes. These 395 stations cover the elevation range of ∼250–4150 m as well as different topographic

positions including peaks, slopes, plains and deep valleys.

All temperature observations were filtered by threshold (ranges from -60 to 60 °C) and visually checked to screen obviously





wrong values. Time offsets between observations and ERA-Interim are avoided by conducting all analyses in UTC time. When using mean daily temperature, days with missing data were removed before further analysis. As a result, there are $\sim 2.5 \times 10^6$ observations of mean daily temperature in or after 1980 used here.

### 3.3 DEM

The fine-scale topography was represented using a DEM with a resolution of 3 arc-second ($\sim$90 m). To avoid the noise in the original dataset, the DEM used in this study was aggregated from the original Global Digital Elevation Model version 2 (GDEM2) with a grid spacing of 1 arc-second (Tachikawa et al., 2011; Meyer et al., 2011) to a spacing of 3 arc-second.

## 4  Methods

Figure 2 shows a flowchart (a) and schematic illustration (b) of REDCAPP. The main steps can be summarized as: (1) obtain
$T_{sa}$ and interpolate $T_{pl}^c$ and $T_{pl}^f$ from the pressure level data (described in Section 4.1); (2) derive $\Delta T^c = T_{sa} - T_{pl}^c$ (described in Section 4.2); (3) estimate $LSCF$ and hence $\Delta T^f$ from the fine-scale DEM (described in Section 4.2); (4) obtain fine-scale $T$ by adding $\Delta T^f$ to $T_{pl}^f$.

The fundamental of REDCAPP is coupling the $\Delta T$ to the $T_{pl}$ at each site and could be given by

$$T = T_{pl} + \Delta T \tag{1}$$

where $T_{pl}$ is the air temperature of pressure level from ERA-Interim, and $\Delta T$ is the influences of land surface. In responding to the required fine scale of $T$, Eq. (1) could be changed to

$$T = T_{pl}^f + \Delta T^f \tag{2}$$

where $T_{pl}^f$ and $\Delta T^f$ is the $T_{pl}$ and $\Delta T$ at the elevation of fine-scale topography.

### 4.1  Interpolation of air temperature

By following Fiddes and Gruber (2014), $T_{pl}^f$ and $T_{pl}^c$ at a given site are obtained by 3D interpolation of $T_{pl}$. This is achieved in two steps: (1) 2D interpolation: derive the elevation of each pressure level by normalizing geopotential height (Eq. 3), and then conduct horizontal 2D interpolation of temperature and elevation for each pressure level; (2) 1D interpolation: vertically interpolate $T_{pl}$ at different heights over one location to the required elevation.

$$Elevation = \frac{\phi}{g_0} \tag{3}$$

where $\phi$ is the geopotential height and $g_0$ is the acceleration due to gravity of $9.80665 \ ms^{-2}$. The geopotential and $T_{pl}$ are extrapolated in the area where the pressure is greater than that of lowest level of ERA-Interim ($\sim$1000 mb) by using values of the lowest two pressure levels. The coarse-scale topography and $T_{sa}$ are bi-linearly interpolated to the resolution of the fine-scale grid in order to avoid blocky artefacts introduced by sudden changes of $\Delta T$ at the boundary of ERA-Interim cells.



## 4.2 Land surface correction factor

The land-surface effect $\Delta T^c$ on simulated near-surface air temperature is given by

$$\Delta T^c = T_{sa} - T^c_{pl} \tag{4}$$

$LSCF$ is introduced here as a scale factor to obtain $\Delta T^f$ from $\Delta T^c$. Therefore, Eq. (2) becomes

$$T^f = T^f_{pl} + LSCF * \Delta T^c \tag{5}$$

where $LSCF$ describes the effect of fine-scale topography on the relative magnitude of land-surface effects. It is parameterized as

$$LSCF = \alpha * h + \beta * v, \tag{6}$$

where $\alpha$, $\beta$ are positive numbers obtained from fitting with observations, and $h$ and $v$ [0,1] are factors derived heuristically
from geomorphology on the fine-scale topography. The lower-case variables of $h$ and $v$ are derived by scaling hypsometric position ($H$) and the "degree of valleyness"($V$) with a scaling factor

$$S = exp(\frac{-R}{\gamma}), \tag{7}$$

where $R$ is the elevation range in a prescribed neighbourhood of analysis and $\gamma$ is a fitting parameter. This scaling reflects the fact that stronger topographic effects on air temperature are to be expected with increasing elevation range. $S$ is equal to one
for $R = 0$ and zero for very large $R$ (Figure 3a).

Hypsometric position $H$, the basis for $h$, is the ratio of the number of cells with higher elevation than a given site to the total number of cells in a prescribed neighbourhood of analysis. It ranges from 1 (deepest valley) to 0 (highest peak). The prescribed neighbourhood of analysis for both $H$ and $R$ is taken as 30 km×30 km. For computational efficiency, $H$ is derived based on a DEM aggregated to 15 arc-second (∼450 m) and the results are nearly identical (Appendix B1). Then, $H$ is scaled to obtain:

$$h = H * (1 - S) + S. \tag{8}$$

The lowest point in the landscape thus always receives a weight of 1 in $h$ (Figure 3b).

The factor $v$ is based on scaling a measure of the "degree of valleyness" [0,1]:

$$v = V * (1 - S), \tag{9}$$

where $v$ becomes larger with increasing elevation range (Figure 3c) and $V$ is described by the normalized MultiResolution
index of Valley Bottom Flatness (MRVBF):

$$V = \frac{MRVBF}{MRVBF_{max}}, \tag{10}$$





where $MRVBF$ identifies valley bottoms occurring at a range of scales (Gallant and Dowling, 2003), and $MRVBF_{max}$ is a constant value of 8 based on the maximum $MRVBF$. The original slope threshold used to scale flatness of topography, is increased to 50% in this study, so that the MRVBF is smoother (Appendix B2).

The main parameters for REDCAPP, denoted by the greek letters $\alpha$, $\beta$, and $\gamma$, are derived from fitting with observational data. For this, values for $LSCF$ were fitted where observations exist. Then, model parameters for predicting these $LSCF$ were derived using global optimization function $differential\_evolution$ of the Python package SciPy (Storn and Price, 1997).

### 4.3 Reference methods

Three reference methods using different sources of air temperature and lapse rate are used to compare with the new downscaling scheme (Table 2, Figure 2). $T_{sa}$ is extrapolated by using a fixed lapse rate of $-6.5$ °C km$^{-1}$ (REF1) and by using variable lapse rate modeled from $T_{pl}$ (REF3) (Blandford et al., 2008; Gao et al., 2012). Linearly interpolated $T_{pl}$ is referenced as REF2 (Fiddes and Gruber, 2014; Gupta and Tarboton, 2016). Since only the upper-air temperatures are used in REF2, this is equivalent to setting $LSCF$ uniformly to 0 (no land-surface influence), while $LSCF$ is uniformly considered to be 1 in REF1 and REF3, which use $T_{sa}$ as their base temperature. To evaluate the performance of REDCAPP against the three reference methods, the coefficient of determination ($R^2$), root mean squared error (RMSE) and mean bias (BIAS), were computed here.

$$RMSE = \sqrt{\frac{\sum_{t=1}^{N}(OBS_t - MOD_t)^2}{N}} \tag{11}$$

$$BIAS = \frac{1}{N}\sum_{t=1}^{N}(MOD_t - OBS_t) \tag{12}$$

### 5 Results

In this section, results are presented in the order of research questions outlined in the introduction. We first investigate $\Delta T$ and whether it can be used for parameterizing cold air pooling and surface effects. Then, we investigate $LSCF$ and its estimation based on a fine-scale DEM. Finally, the performance of REDCAPP is evaluated.

### 5.1 Properties of $\Delta T$

Figure 4 presents seasonal variations of daily $\Delta T^c$. In general, $\Delta T^c$ is close to 0 °C in warm seasons with the median value sightly above 0 °C in the Swiss Alps from March to June and greater than -0.8 °C in the Qilian Mountains from April to June. In winter, lower median $\Delta T^c$ values are found in both the Swiss Alps and the Qilian Mountains. Furthermore, a larger range of $\Delta T^c$ in winter is caused by lower minima of $\Delta T^c$, likely related to radiative cooling.

Figure 5 shows one year of daily $\Delta T^c$ as well as $T$ derived from observation and downscaling at selected sites. The downscaled series are either ignoring $\Delta T^c$ (REF2) or adding it uniformly (REF3) to all stations. Daily $\Delta T^c$ shows a similar pattern as Figure 4. At the mountain sites (COV, BEV1, DDS; see Table 3), $T_{pl}^f$ describes $T_{obs}$ well without accounting for $\Delta T^c$ (REF2), and the RMSEs were less than 1.4 °C (Table 3). By contrast, REF2 does not describe $T_{obs}$ well at valley locations,





especially in winter, and RMSEs are markedly higher. In comparison, the results of REF3, through adding $\Delta T^c$ to $T_{pl}^f$, follow $T_{obs}$ better at valley sites (SAM, SIA, EBO) and worse at mountain sites. Although REF3 improves predictions in deep valleys (e.g. SAM), results in winter are still higher than the observations because winter inversions here are stronger than predicted by $\Delta T^c$.

These results highlight the spatial and temporal variability of land-surface effects on $T$. As the full incorporation of $\Delta T^c$ in downscaling improves predictions in valley locations and degrades them in mountain sites, a spatially-variable $LSCF$ appears to be a promising means for better predicting land-surface effects on $T$ at the fine scale.

## 5.2   Land Surface Correction Factor

At the example of selected stations (Table 3), Figure 6 shows that $\Delta T^c$ correlates with the difference of observed temperature
and a prediction involving pressure levels, only ($T_{obs} - T_{pl}^f$). Therefore, $\Delta T^c$ can be used to correct for some of the difference found between them. The fitted $LSCF$ is related to the topography and increases from near 0 at mountain peaks to almost 2 in deep valleys. This indicates the possibility of predicting $LSCF$ based on a DEM. Furthermore, fitted $LSCF > 0$ hint at the possibility of representing CAPs by using a $LSCF$ and $\Delta T^c$.

To assess the performance of DEM-derived $LSCF$ (based on Eq. 5), we conducted a ten-fold cross-validation, separately for
the Swiss Alps and the Qilian Mountains (Figure 7). Each time, $\sim$90 % of the observations are randomly selected for deriving model parameters and the remaining 10 % are used for evaluation. Results show an RMSE of 0.29 and 0.26, a BIAS of 0 and 0.03 as well as a $R^2$ of 0.69 and 0.60 in the Swiss Alps and the Qilian Mountains. These results indicate that $LSCF$ can be estimated from a DEM based on geomorphometry and that results will be useful in improving downscaling.

Model parameters for estimating $LSCF$ were derived by using all stations but separately for the Swiss Alps and the Qilian
Mountains (Table 4). Figure 8 shows the spatial fields of topographic factors in selected area based on the modeled factors. Hypsometric position and normalized MRVBF, and therefore $LSCF$, vary strongly with topography. In the test area shown, $LSCF$ ranges from near 0 on mountain peaks to about 1.67 in deep valleys.

## 5.3   Performance of REDCAPP

### 5.3.1   Comparison with station data

Figure 9 shows plots of $T_{obs}$ against results of REF1, REF2, REF3 and REDCAPP (MOD) and indicates that REDCAPP improves the prediction of $T$ over reference methods. Nearly all measures of agreement or deviance improve with REDCAPP when compared to reference methods, with the exception of an increased BIAS in the Swiss Alps when compared to REF3.

Figure 10 shows the seasonal deviance of downscaled daily results (MOD - OBS) for different methods. Similar to the detailed comparison of typical stations showed in Figure 5, REF2 captures temperatures in summer well but has a warm bias
in winter. By contrast, REF1 predicts $T$ too low in winter. This is because the lapse rates are expected to decrease owing to the presence of CAPs. There is no obvious seasonal trend in the median deviation of REF3. However, the minimum of deviation is smaller than REF2 in winter. REDCAPP captures $T$ well in both winter and summer. The median deviation for each month



was within ± 0.50 °C (from -0.06 to 0.48 °C) in the Swiss Alps and within ± 0.55 °C (from -0.53 to 0.45 °C) in the Qilian Mountains.

Figure 11 shows the deviances of downscaled results by elevation. REF2 performs well at high elevation areas with the median deviance close to 0, but has a warm bias with decreasing elevation. By contrast, REF1 and REF3 tend to have a cold

bias at high elevation and often an increasing range of deviance with elevation. REDCAPP captures $T$ well across elevations. The median deviance was within ± 0.70 (from -0.24 to 0.68) °C in the Swiss Alps and within ± 1.25 (from -0.76 to 1.22) °C in the Qilian Mountains.

Figure 12 shows a comparison of REF2, REF3 and REDCAPP with time series at selected sites. Similar to the fitted LSCFs, DEM-derived LSCFs are spatially variable and increase from near 0 near mountain peaks to more than 1 in some slopes and

deep valleys. REDCAPP improves the prediction of $T$ at all the topographic positions by comparing with reference methods. In summer, REDCAPP captures $T$ well, in winter, the BIAS is decreased through adding the influences of CAPs, especially by using the DEM-derived LSCF larger than 1.

### 5.3.2  Spatial signature of REDCAPP

Figure 13 shows the spatial variation of mean annual $\Delta T^f$ for the year 2015. In valleys, downscaled $T$ can be up to -2.1 °C

lower than $T_{pl}^f$. With increasing elevation, the simulated land-surface effect decreased to almost 0 °C. This gives a clear picture on the topography-related spatial variability of $\Delta T^f$ and indicates REDCAPP can capture the variations well.

## 6  Discussion

In this section, we discuss advantages and limitations of the model, and how it could be further refined in the future. We have demonstrated that information from coarse-scale models ($\Delta T^c$) can be used as a proxy of land-surface effects and with

a disaggregation factor ($LSCF$) estimated from a fine-scale DEM can improve air-temperature downscaling in mountains. At the same time, this finding needs to be put into perspective: a full simulation of the atmospheric physics and land surface at high resolution will likely outperform this parameterization but at a cost that is orders of magnitude higher (Fowler et al., 2007). Ultimately, the choice of method (or combination of several methods) depends on the problem at hand. It is likely that the parameterization put forward here can be further improved in its ability to predict fine-scale patterns and its suitability for

transferring parameters between areas and thus, the suitability for application in data-sparse regions. Nevertheless, REDCAPP and similar methods (Fiddes et al., 2015; Gupta and Tarboton, 2016) demonstrate that coarse-scale information on atmospheric variables can contribute to better prediction at finer scales without the need for increased resolution in the atmospheric model.

### 6.1  Land surface correction factor

For simplicity, we model the influence of CAP and other land-surface effects on $T$ with one $LSCF$ that varies spatially but

is constant over time. This lumped nature of $LSCF$ is imperfect because the presence of strong valley inversion in winter and their absence in the warm season would suggest a seasonally variable $LSCF$. In other words, $LSCF$ is expected to be



greater in winter than in summer as the fractional influence of CAP (the part of $\beta$ in Eq. 5) should be removed from $LSCF$. In REDCAPP, applying the same $LSCF$ year-round to $\Delta T^c$ will make downscaled $T$ higher in winter and lower in summer. A potential avenue for addressing this problem is simulating the likelihood for CAPs based on surface net radiation or Richardson number based on the reanalysis data.

## 6.2 Transferability

Based on the ten-fold cross-validation, $LSCF$ is modeled well in both the Swiss Alps and the Qilian Mountains. The resulting parameter values, however, are different (Table 4). The reasons for this can be speculated to include differences in topography (e.g. valley shape), the numbers and distribution of stations used, climate (continentality, effect of lumping two processes into one $LSCF$), or differences in land-surface characteristics (e.g. canopy and snow cover). The difference in estimated parameter values limits the transferability of REDCAPP as it requires tests new mountain regions to investigate suitability. This is a significant drawback and we hope that over time, application in many mountain ranges will help to establish correlations of trusted parameter values with environmental conditions.

## 6.3 Input data

Although we only apply and test our method with ERA-Interim here, it can be used with other reanalyses such as CFSR, NCEP, MERRA or 20CRV2. Besides global reanalyses, regional high-resolution assimilations produced by RCMs (e.g. E-OBS, Chinese Academy of Sciences forcing data, ASR) (Chen et al., 2011), and upper-air temperature reanalyses (e.g. ASR) may be suitable alternatives in some regions. These regional assimilations often capture surface air temperature better by assimilating more observations and by using finer grids than global reanalyses. Since upper-air temperature and $\Delta T$ are treated separately in REDCAPP, they can also be derived from different data sources.

## 6.4 Future development

After lapse-rate correction, the key issue for $T$ downscaling (not only in mountain regions) is resolving variations caused by the variable land surface (e.g. elevation, heating/cooling, CAP). This method proposed here allows predicting land surface influences on $T$ as a function of topography (like we did here) and it can potentially be extended to include other surface conditions (e.g. snow, canopy, soil moisture), which are considered important (Lin et al., 2016; Liston and Elder, 2006).

## 7 Conclusions

We describe and test a downscaling method for near-surface air temperature. It derives $\Delta T$ from coarse-scale atmospheric model data as a proxy of the effect that the land-surface has on near-surface air temperature. The magnitude of this effect is adjusted at the fine scale based on geomorphometric characteristics derived from a fine scale of DEM. The results from the new method are evaluated with 395 stations in two mountain ranges, leading to these conclusions: (1) The proxy $\Delta T$ is suitable for parameterizing CAP and surface effects. (2) The land surface correction factor $LSCF$ can be predicted from a fine-scale



DEM where $\sim 70\%$ of the variance in directly fitted $LSCF$ could be explained by the parameterization. (3) REDCAPP improves downscaling when compared with reference methods. This is primarily because the advantages of REF2 and REF3 are combined. (4) The transfer of REDCAPP parameters between mountain ranges is difficult and at present, separate fitting parameters in new regions is recommended.

5    REDCAPP can produce daily, high-resolution (here $\sim$90 m) gridded fields of near-surface air temperature in mountains. This can provide input for other models simulating phenomena related to e.g., hydrology, permafrost, ecology. The input data are not limited to ERA-Interim and could be extended to other reanalyses such as CFST, NCEP, MERRA or 20CRV2.

## 8    Data and code availability

10    REDCAPP, written in Python, is included as supplement and updates will be available via GitHub (https://github.com/geocryology/REDCAPP). The observation and reanalysis data used are not openly available online, but for research purposes can be requested from the sources cited.

## Appendix A:  Nomenclature

In this study, $T$ refers to air temperatures, subscripts identify the source (obs: observation; sa: surface analysis; pl: pressure level), superscripts identify at which elevation this is (c: coarse-scale, f: fine-scale). Coarse-scale elevation refers to the topography used by the re-analysis, fine-scale refers to the DEM used for downscaling. Observations refer to mean daily air temperature and are assumed to be at the elevation of fine-scale topography. The surface analysis fields in the reanalysis are given at the elevation of the coarse scale topography (as obtained from invariant geopotential ERA-Interim file by Eq. 3), pl represents air temperature derived from pressure levels and can be either. Table A1 provides list of symbols.

## Appendix B:  Topography factors

### B1    Hypsometric position

To improve computational effectiveness, hypsometric position is derived by using a lower-resolution DEM (15 arc-second or $\sim$450 m) derived by aggregating the fine-scale DEM. Figure A1 presents the comparison of hypsometric position obtained from original fine-scale (3 arc-second or $\sim$90 m) and aggregated DEMs in the Swiss Alps and in the Qilian Mountains.

### B2    Multiresolution index of valley bottom flatness index

The choice of suitable parameters for MRVBF is affected by the resolution of the input DEM as well as different landscape characteristics and applications (Gallant and Dowling, 2003). In responding the cold air pooling movement, the slope threshold is here adjusted from the original value of 16% to 50% in this study. Figure A2 compares MRVBF using slope threshold of 50%





and 16% (original paper) in the Alps and the Qilian Mountains. The results indicate that MRVBF is smoother when using the larger threshold and hence likely describes cold air movement better. The threshold value of 50 was chosen after considerable tests and comparisons but ultimately remains a subjective choice at this time.

*Author contributions.* Bin Cao carried out this study by analyzing data, developing most of the model code, performing the simulations and by structuring as well as writing the paper. Stephan Gruber conceived and guided the project, designed part of the code and contributed to structuring and writing the paper. Tingjun Zhang contributed to the writing of the paper.

*Acknowledgements.* We would like to express our gratitude to John C. Gallant for his help with the Multiresolution Index of Valley Bottom Flatness. Station data in Switzerland is provided by the Swiss Federal Office of Meteorology and Climatology (MeteoSwiss) and Inter-cantonal Measurement and Information System (IMIS) from the WSL Institute for Snow and Avalanche Research SLF. The authors would like to thank Dr. Joel Caduff for help with the IMIS dataset. We would also like to thank the Heihe Watershed Allied Telemetry Experimental Research (HIWATER) project and Third Pole Environment Database for providing air temperature in the Qilian Mountains. We thank ECMWF for the ERA-Interim reanalysis data. This study was supported by the National Natural Science Foundation of China (91325202), the National Key Scientific Research Program of China (2013CBA01802), partly by the Fundamental Research Funds for the Central Universities (China), and by the projected of "Quantifying the Hidden Thaw" funded by the Canada Foundation for Innovation (CFI). The ASTER dataset are downloaded from website http://gdex.cr.usgs.gov/gdex/.



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



**Table 1.** Summary of observational stations used.

| Region | Data Source | Number of sites |
|---|---|---|
| Swiss Alps | MeteoSwiss | 184 |
| | IMIS | 178 |
| Qilian Mountains | HIWATER | 30 |
| | TPED | 3 |





**Table 2.** Summary of reference methods.

| Reference Method | Lapse Rate | Base Temperature | $LSCF$ |
|---|---|---|---|
| REF1 | $-6.5\,°\mathrm{C}\ \mathrm{km}^{-1}$ (fixed) | $T_{sa}$ | 1 |
| REF2 | $T_{pl}$-based (variable) | $T_{pl}$ | 0 |
| REF3 | $T_{pl}$-based (variable) | $T_{sa}$ | 1 |

REF2 is from Fiddes and Gruber (2014), while REF3 is from Gao et al. (2012)



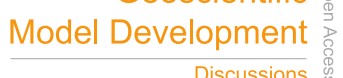

**Table 3.** Comparison of observations against reference methods of REF2 and REF3.

| Station | Location | | | | REF2 | | REF3 | |
|---------|----------|----------|---------|------------|------|------|------|------|
| | lat (°) | lon (°) | ele (m) | Topography | RMSE | BIAS | RMSE | BIAS |
| COV | 46.4180 | 9.8212 | 3351 | Peak | 1.01 | 0.24 | 1.63 | -0.41 |
| DDS | 38.0142 | 100.2421 | 4147 | Peak | 1.34 | 0.56 | 2.38 | -1.05 |
| BEV1 | 46.5487 | 9.8538 | 2490 | Peak | 1.22 | 0.54 | 1.41 | 0.04 |
| EBO | 37.9492 | 100.9151 | 3294 | Slope | 3.31 | 2.41 | 1.87 | 0.43 |
| SIA | 46.4323 | 9.7623 | 1853 | Valley | 2.44 | 1.14 | 1.65 | 0.50 |
| SAM | 46.5263 | 9.8789 | 1756 | Valley | 3.85 | 1.95 | 2.81 | 1.39 |

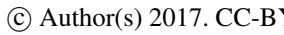



**Table 4.** Summary of model parameters for estimating $LSCF$ from DEMs.

| Area | Model parameters | | | Evaluation | | |
|---|---|---|---|---|---|---|
| | $\alpha$ | $\beta$ | $\gamma$ | $R^2$ | RMSE | BIAS |
| Swiss Alps | $0.61 \pm 0.03$ | $1.56 \pm 0.04$ | $465 \pm 50$ | 0.69 | 0.29 | 0.00 |
| Qilian Mountains | $0.90 \pm 0.08$ | $0.34 \pm 0.11$ | $138 \pm 20$ | 0.68 | 0.26 | 0.03 |

The value after $\pm$ are standard deviation derived from ten-fold cross-validation.





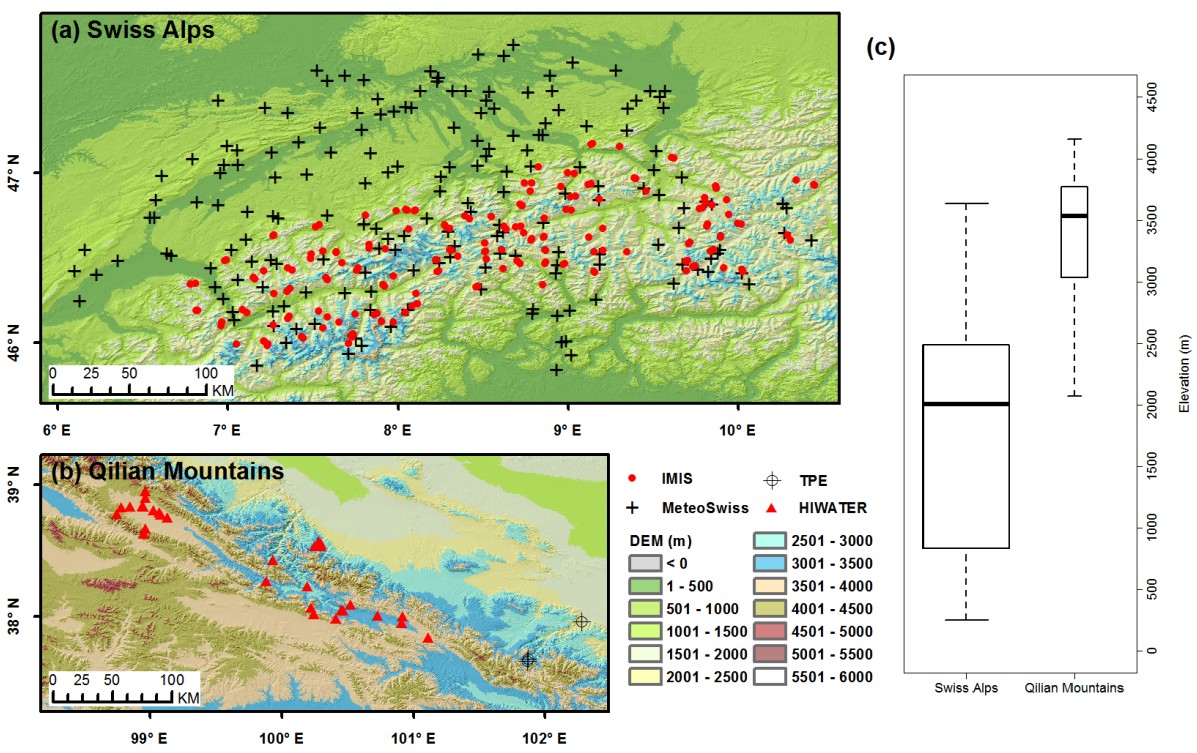

**Figure 1.** Experimental region of and observation stations in the Swiss Alps (a) and the Qilian Mountains (b). Elevation distribution of observation station (c). N is the number of observation stations.



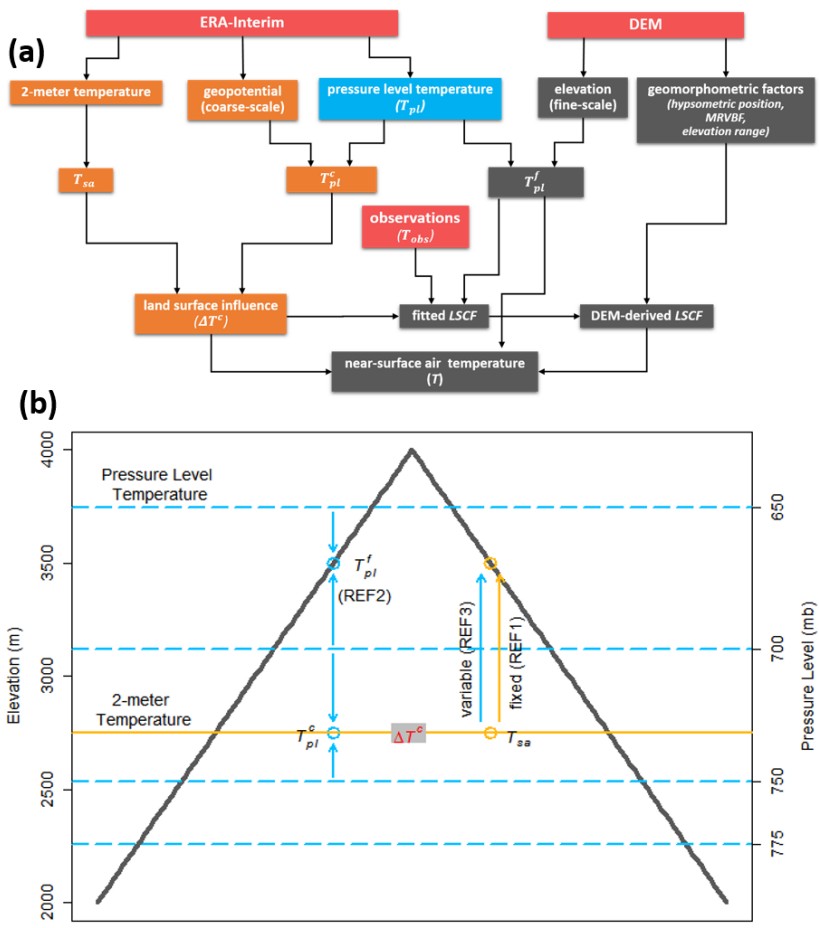

**Figure 2.** Model flow chart (a) and schematic illustration of interpolations and reference methods (b). Red squares denote input datasets. Variables at the elevation of coarse-scale topography are marked in yellow while the elevation of fine-scale topography is shown in grey. $T_{pl}$ in blue could be at both coarse and fine scale of elevation. Blue arrows and points are variable lapse rates and temperatures derived from $T_{pl}$, while yellow point is temperature derived from $T_{sa}$ and yellow arrow is the fixed lapse rate of $-6.5\ °C\ km^{-1}$. Detailed symbol and variable names can be found in Appendix A. Schematic illustration is revised from Fiddes and Gruber (2014).





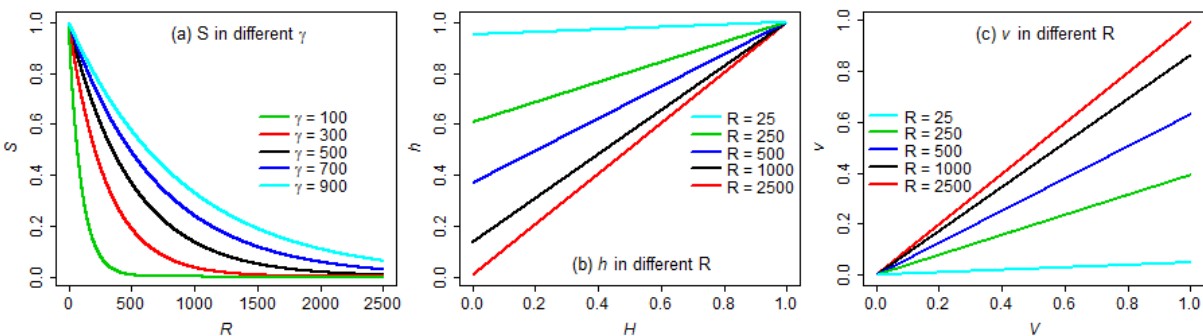

**Figure 3.** (a) Scale factor ($S$) decreases with increasing elevation range ($R$) by using different $\gamma$ values. Fraction influence of surface effect on $T$ ($h$) increases with hypsometric position ($H$) and strength of CAP ($v$) increases with normalized MRVBF ($V$). $\gamma$ is 500 in figure (b) and (c). The lowest point in the landscape always receive a weight of 1 in $h$ and a weight of 0 in $v$.




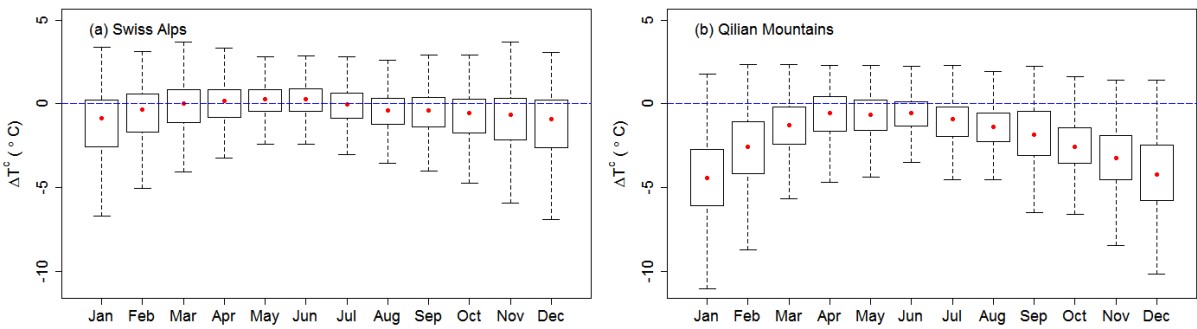

**Figure 4.** Seasonal changes shown as monthly distributions of average daily $\Delta T^c$ derived from ERA-Interim for the locations of all stations.



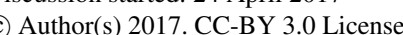


**Figure 5.** Detailed timeseries of $\Delta T^c$, $T_{obs}$, REF2 and REF3 at the selected stations from different geomorphometric positions.



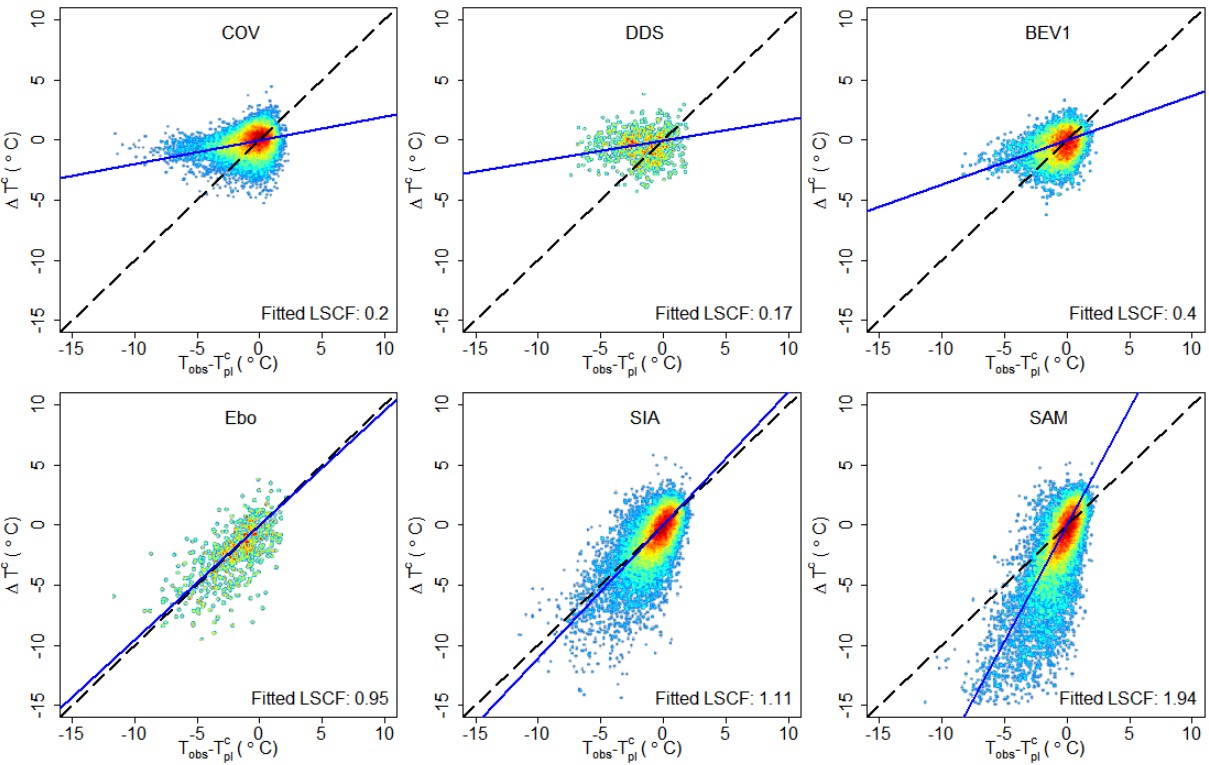

**Figure 6.** Difference of observed temperature and a prediction involving pressure levels ($T_{obs}$-$T_{pl}^{f}$) against $\Delta T^{c}$. The representation is a smoothed color density of a scatter plot to make a quality of points visual. The lines are results of $LSCF * \Delta T^{c}$ by using $LSCF = 1$ (black dash) and best-fitted $LSCF$ (blue solid) at the selected stations.



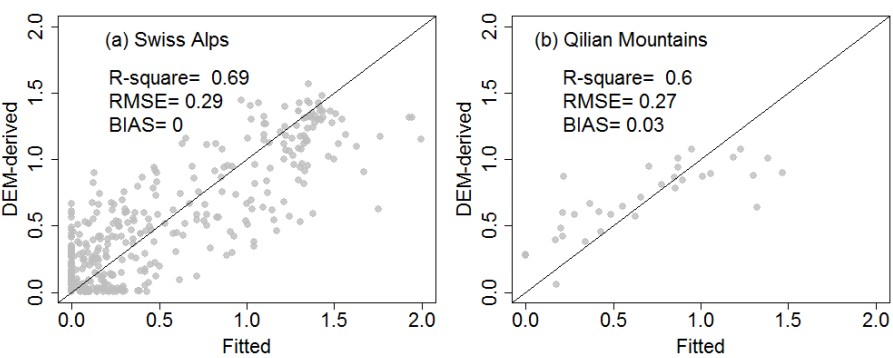

**Figure 7.** Ten-fold cross-validation of DEM-derived $LSCF$ against fitted values in the Swiss Alps (a) and the Qilian Mountains (b).



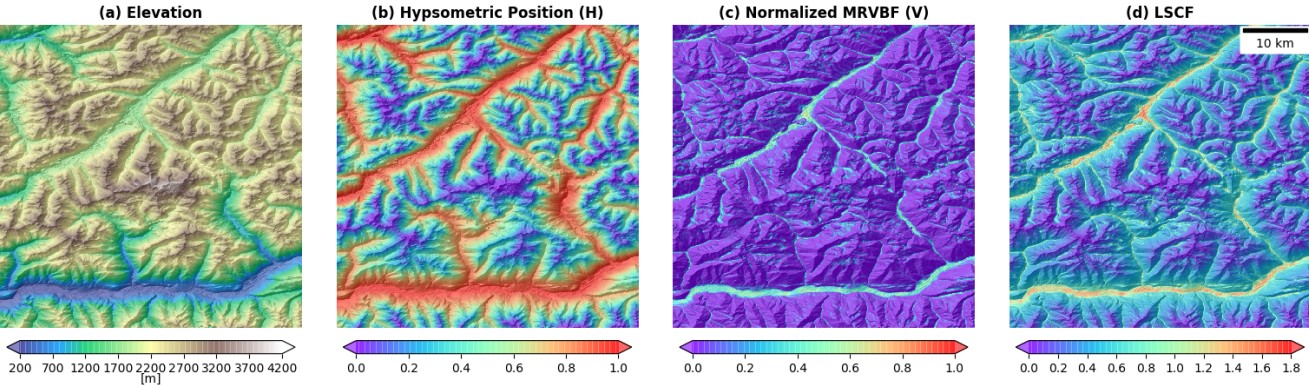

**Figure 8.** Spatial variation of elevation (a), hypsometric position (b), normalized MRVBF (c) and $LSCF$ (d) in selected slope terrain (9.2999° N, 10.3998° W, 46.9999° N, 45.9999° S).





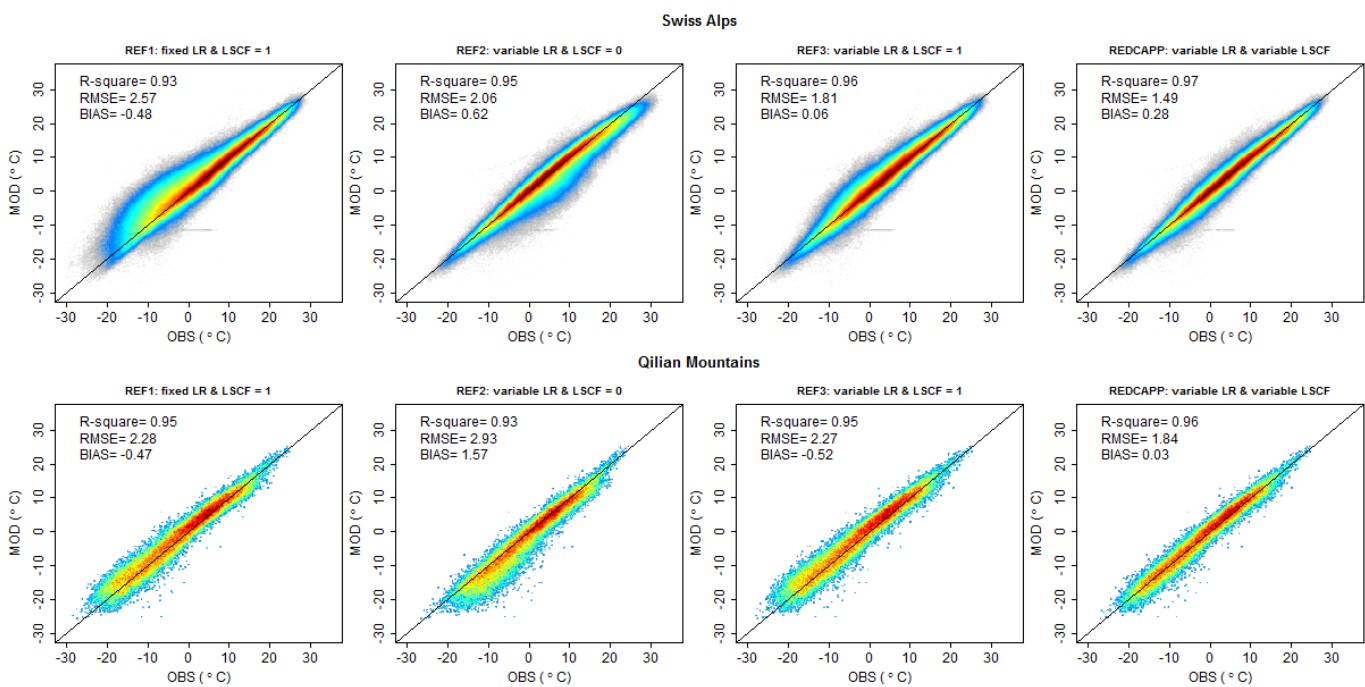

**Figure 9.** $T_{obs}$ (OBS) against results of REF1, REF2, REF3 and REDCAPP (MOD). LR in subtitle means lapse rate.




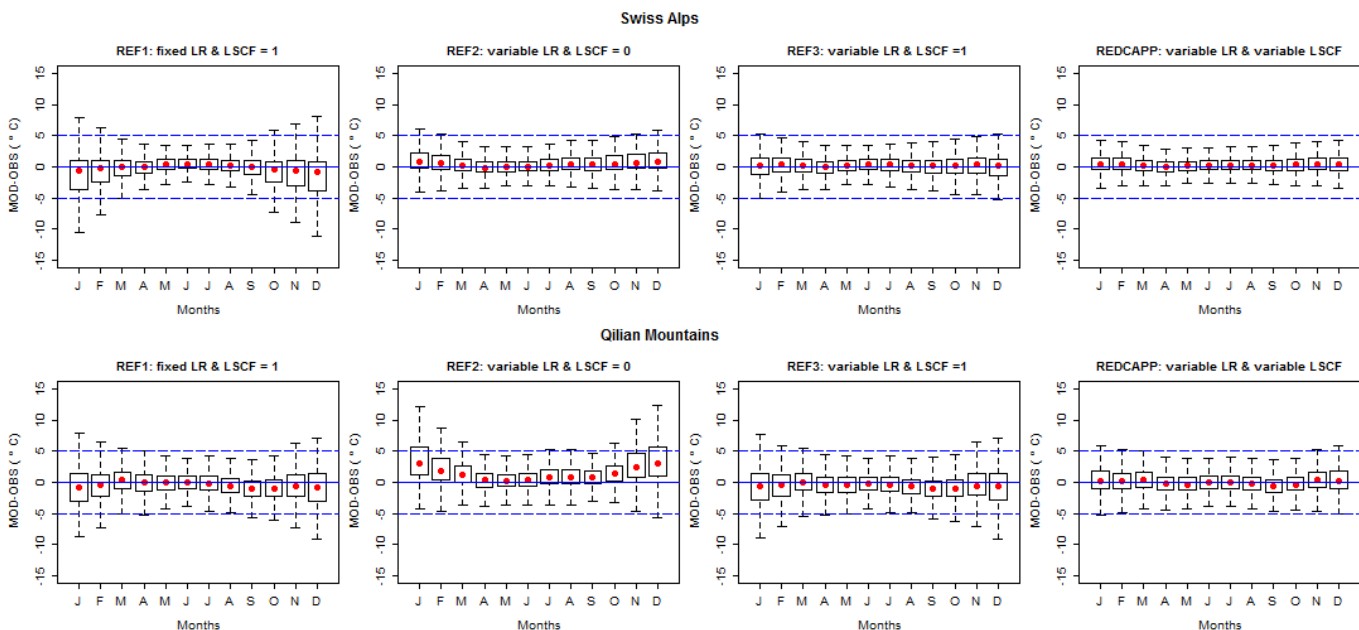

**Figure 10.** Seasonal deviance of downscaled daily results (MOD-OBS) for different methods.



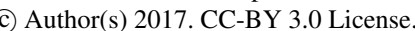

**Figure 11.** Deviances of downscaled results by elevations. The stations are grouped by elevation with an interval of 300 m. Each box may contain multiple stations and the numbers of observation times (days) are given in blue on the right.





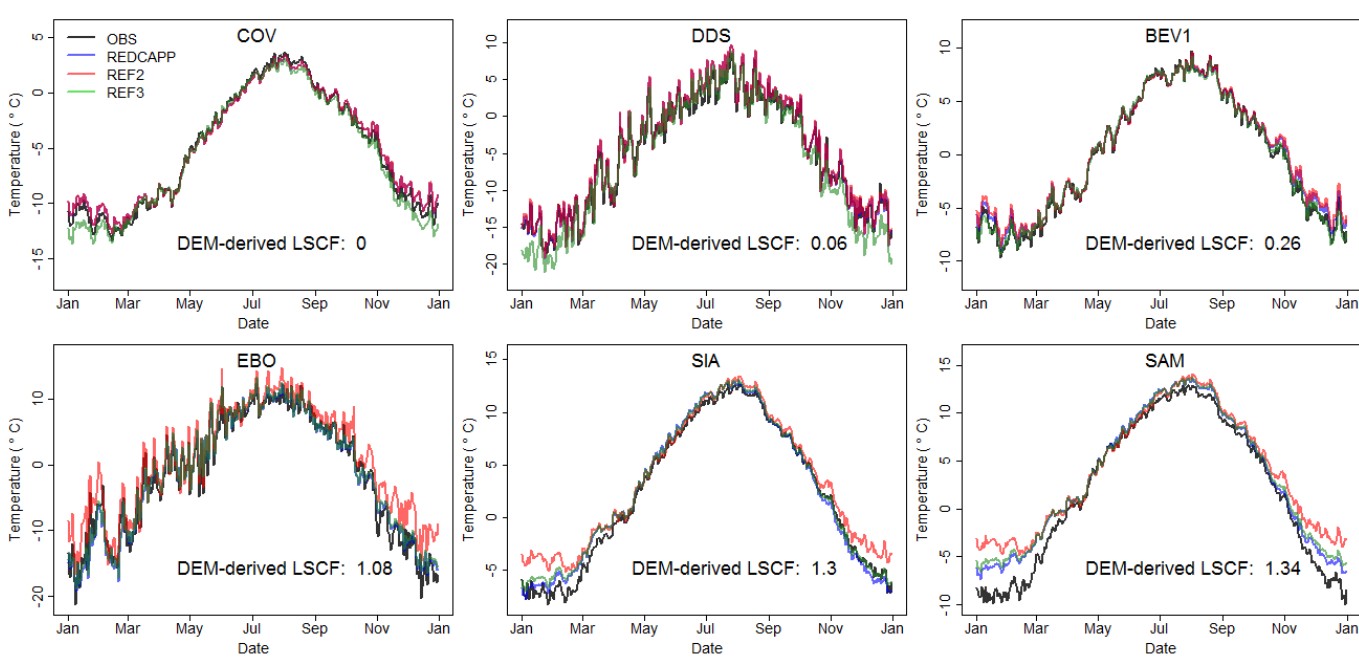

**Figure 12.** Comparison of REF2, REF3 and REDCAPP with time series at selected stations. The daily temperatures present are averaged based on all available years; the shorter time series for EBO explains the larger variation in the plot.





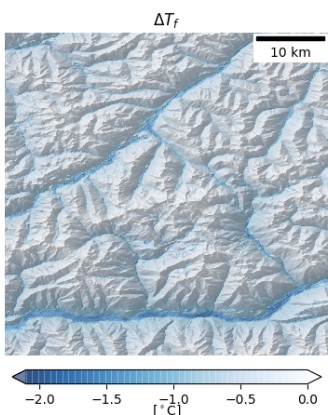

**Figure 13.** Fine-scale of land-surface influence ($\Delta T^f$) for the test area (9.2999° N, 10.3998° W, 46.9999° N, 45.9999° S).





**Table A1.** Table of symbol and variables for REDCAPP

| Symbol | Name | Unit |
|---|---|---|
| $T$ | Near-surface air temperature | °C |
| $T_{obs}$ | Observational surface air temperature | °C |
| $T_{sa}$ | 2–meter air temperature at the elevation of coarse scale | °C |
| $T_{pl}$ | Air temperature of pressure level in reanalysis, known as upper-air temperature | °C |
| $T_{pl}^{c}$ | Air temperature of pressure level at the elevation of coarse scale | °C |
| $T_{pl}^{f}$ | Air temperature of pressure level at the elevation of fine scale | °C |
| $\Delta T$ | Land-surface influences on surface air temperature | °C |
| $\Delta T^{c}$ | Land-surface influences on surface air temperature at elevation of coarse scale | °C |
| $\Delta T^{f}$ | Land-surface influences on surface air temperature at elevation of fine scale | °C |
| $LSCF$ | Land surface correction factor | – |
| $\alpha$ | Fractional influence of surface effects on air temperature | – |
| $\beta$ | Influences of cold air pooling on air temperature | – |
| $h$ | Scaled hyposmetric position | – |
| $H$ | Hypsometric position | – |
| $v$ | degree of valleyness | – |
| $MRVBF$ | Multiresolution index of valley bottom flatness | – |
| $V$ | Normalized multiresolution index of valley bottom flatness | – |
| $R$ | Elevation range in a prescribed neighbourhood | m |



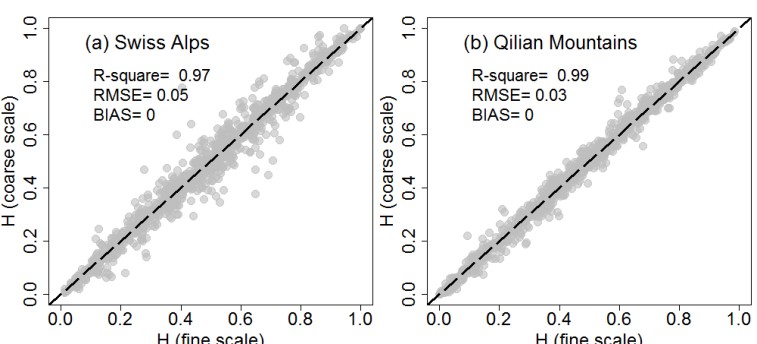

**Figure A1.** Comparison of hypsometric position derived from coarse-scale (15 arc-second) and fine-scale (3 arc-second) DEMs based on a random sample of 1000 points.





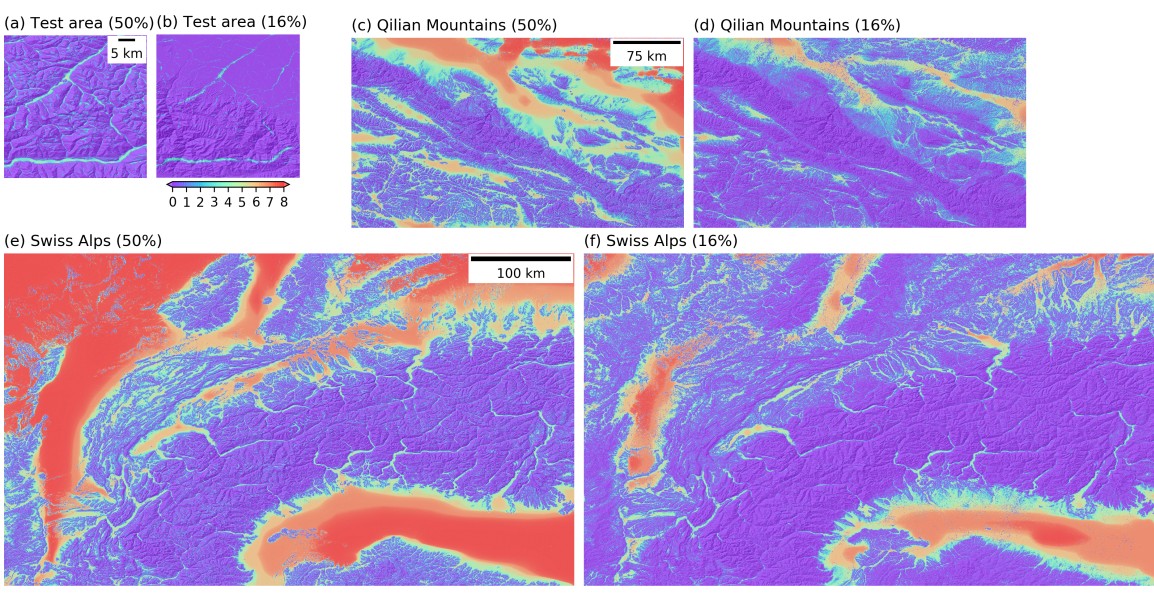

**Figure A2.** Original MRVBF in the test area, Swiss Alps and the Qilian Mountains by using slope threshold of 50% and 16%.