# Peer review of "REDCAPP (v1.0): Parameterizing valley inversions in air temperature data downscaled from re-analyses"

_Geoscientific Model Development, 2017_

## Referee Comment (RC1) · Anonymous Referee #1 · 17 May 2017

The manuscript "REDCAPP (v1.0): Parameterizing valley inversions in air temperature data downscaled from re-analyses" by Cao et al. presents a new technique to downscale temperature data in mountainous regions whereby they develop a land-surface correction factor. They then demonstrate this technique for two mountainous regions: the Swiss Alps and Qilian Mountains. The technique that the authors develop is of interest to GMD readership, but the following comments need to be addressed before the manuscript is considered for publication.

General comments:

I am missing a discussion of how the method that the authors developed in the present manuscript differs from other downscaling approaches that already exist for complex or

mountainous terrain. There needs to be more discussion of how the authors' method improves upon and is better than pre-existing downscaling techniques that also use variable lapse rates and incorporate information about the land surface and topographical characteristics, e.g. the Parameter-elevation Regression on Independent Slopes Model (PRISM) (Daly et al., 2000; Daly et al., 2002), the Daily Surface Weather and Climatological Summary (DAYMET) (e.g., Thornton et al., 1997), and the techniques used in Hijmans et al. (2005).

The temperature lapse rate is defined as decreasing with height and thus a negative lapse rate implies a temperature increase with height. This change needs to be implemented throughout the manuscript.

In Section 3.2, more discussion is required about the meteorological stations that the authors used, e.g., instrument type, completeness of the data sets at these stations within the two study regions, etc. Also, the authors mention in line 3 of page 5 that mean daily temperatures in 1980 or after are used. The total time period of the study needs to be indicated. I am also not sure what the authors mean by "obviously wrong values" on pages 4-5 in this section. More description is necessary here too.

In Section 5.3.1, the authors note that the bias in the Swiss Alps increases with the implementation of the REDCAPP technique, but no explanation is offered as to why this is.

The quality of many of the figures needs to be improved. Figure 1 would benefit from a zoomed out map showing the relative locations of the study areas in the Swiss Alps and Qilian Mountains. In Figures 4, 10, and 11, are these means or medians shown with the red dots? Please include this information in the legends for these respective figures. In Figure 5 and in Figure 11, what time period is being shown for each of the stations? This information should to be included either in a separate table or in the figure captions. Finally, the latitude and longitude should be included directly on Figure 8, Figure 13, and Figure 2, rather than in the caption, in order to improve the figures'

readability.

Specific comments:

Page 1, Line 18: Change "oder" to order.

Page 6, Line 7: Include citation for "degree of valleyness."

Page 13: Many of the references are missing doi numbers. Please include these.

Page 25: I am unsure what you mean by "a quality of points visual."

References

Daly, C. G. H. Taylor, W. P. Gibson, T. W. Parzybok, G. L. Johnson, and P. A. Pasteris, 2000: High-quality spatial climate data sets for the United States and beyond. Transactions of the American Society of Agricultural Engineers, 43 (6), 1957-1962.

Daly, C. W. P. Gibson, G. H. Taylor, G. L. Johnson, and P. Pasteris: 2002: A knowledge-based approach to the statistical mapping of climate. Climate Research, 22 (2), 99-113.

Hijmans, R., S. E. Cameron, J. L. Parra, P. G. Jones, and A. Jarvis, 2005: Very high resolution interpolated climate surfaces for global land areas. International Journal of Climatology, 25, 1965-1978. Thornton, P. E., S. W. Running, and M. A. White, 1997: Generating surfaces of daily meteorological variables over large regions of complex terrain. Journal of Hydrology, 190 (3-4), 214-251.

—————————————————————

---

## Author Comment (AC1) · 31 May 2017

**Response to Anonymous Referee #1**

The manuscript "REDCAPP (v1.0): Parameterizing valley inversions in air temperature data downscaled from re-analyses" by Cao et al. presents a new technique to downscale temperature data in mountainous regions whereby they develop a land-surface correction factor. They then demonstrate this technique for two mountainous regions: the Swiss Alps and Qilian Mountains. The technique that the authors develop is of interest to GMD readership, but the following comments need to be addressed before the manuscript is considered for publication.

The authors would like to thank the reviewer for the constructive feedback, and the thorough assessment of the manuscript. Below we provide a point-to-point response to each comment, reviewer comments are given in black, responses are given in blue. Additionally, we have included details of how we intend to address these changes in a revised submission.

**General comments:**

I am missing a discussion of how the method that the authors developed in the present manuscript differs from other downscaling approaches that already exist for complex or mountainous terrain. There needs to be more discussion of how the authors' method improves upon and is better than pre-existing downscaling techniques that also use variable lapse rates and incorporate information about the land surface and topographical characteristics, e.g. the Parameter-elevation Regression on Independent Slopes Model (PRISM) (Daly et al., 2000; Daly et al., 2002), the Daily Surface Weather and Climatological Summary (DAYMET) (e.g., Thornton et al., 1997), and the techniques used in Hijmans et al. (2005).

We will add a detailed discussion (see below) of comparisons of REDCAPP and existing methods (e.g. PRISM, DAYMET and other approaches) as a new Section 6.1 "Comparison with other downscaling techniques".

6.1 Comparison with other downscaling techniques

*Many existing downscaling approaches for mountainous terrain focus on deriving fine-scale T through interpolation (e.g. truncated Gaussian weighting filter, Inverse Distance Weighting, or Kriging) of surrounding observations, and adjustments are then made based on fine-scale topography. PRISM (Parameter-elevation Regressions on Independent Slopes Model) (Daly et al. 2000; Daly et al. 2002), for example, derives a weighing function to represent the relationship of T with geographic (e.g. slopes, coastal) and meteorological (e.g. atmosphere boundary-layer) factors. Similarly, the approach by Thornton et al. (1997) calculates interpolation weights for the stations nearby, and corrected the downscaled results based on an empirical relationship of T to elevation, and Hijmans et al. (2005) conducted a second-order spline interpolation using latitude, longitude and elevation as independent variables. As observations are usually sparse in mountains, especially at higher elevation, these methods are expected to often have significant uncertainty caused by inadequately sampling of elevation and hence lapse rate. In comparison, REDCAPP relies on reanalysis data for air temperature and uses station data only for calibration of the LSCF related to CAP. REDCAPP derives lapse rates from multiple layers of upper air temperature*

*encompassing the entire elevation range of study area. Thus, REDCAPP results are expected to be robust because both the Tsa and Tpl from reanalysis are used.*

The temperature lapse rate is defined as decreasing with height and thus a negative lapse rate implies a temperature increase with height. This change needs to be implemented throughout the manuscript.

Thank you for pointing this out. We corrected throughout the manuscript, changes are listed below:

(1) *"For example, Lewkowicz and Bonnaventure (2011) reported that average lapse rates could be positive in mountains due to strong winter inversion and result in lower T in valleys than at higher locations."* is changed to

*For example, Lewkowicz and Bonnaventure (2011) reported that T in valleys was lower than at higher locations in mountains due to strong winter inversion.*

(2) *"This is because the lapse rates are expected to decrease owing to the presence of CAPs."* is changed to

*"This is because the lapse rates are expected to **increase** owing to the presence of CAPs."*

In Section 3.2, more discussion is required about the meteorological stations that the authors used, e.g., instrument type, completeness of the data sets at these stations within the two study regions, etc. Also, the authors mention in line 3 of page 5 that mean daily temperatures in 1980 or after are used. The total time period of the study needs to be indicated. I am also not sure what the authors mean by "obviously wrong values" on pages 4-5 in this section. More description is necessary here too.

Yes, we agree the discussion of observations will benefit from more detail. We reformulated this part by adding the instrument type and accuracy. Additionally, a figure showing observation completeness is added in Figure 2.

The *"obviously wrong values"* means the value out of the range of -60 to 60 °C or not consistent by comparing values with the day before and after the checking day.

*"The temperature from MeteoSwiss is observed using the Thygan instrument which has an accuracy of ± 0.01 °C, and temperatures from IMIS are measured by several different sensors (including Rotronic MP100H, Rotronic MP102H/HC2, Rotronic MP103A, Campbell Scientific CS215), with sensor accuracies ranging from ± 0.1 to ± 0.9 °C. In the Qilian Mountains, temperature sensor HMP155 with a typical accuracy of ± 0.2 °C are used. The 395 stations used cover an elevation range of ~250–4150 m as well as different topographic positions including peaks, slopes, plains and deep valleys (Figure 2a).*

*All temperature observations were filtered using a threshold (plausible values from -60 to 60 °C), and the outliers of temperature time series were removed by visually check. Time offsets between observations and ERA-Interim are avoided by conducting all analyses in UTC time. When using mean daily temperature, days with missing data were removed before further analysis. Though there are totally 395 stations used here, not all of them are available in a single year (Figure 2b). In total, there are ~2.5 × 10$^6$ observations of mean daily temperature in or after 1980 used here."*

In Section 5.3.1, the authors note that the bias in the Swiss Alps increases with the implementation of the REDCAPP technique, but no explanation is offered as to why this is.

To clarify, we added

*"This is because REF3 resulted in air temperatures being too low at high elevation, while the influence of CAP was underestimated in valleys by applying a fixed LSCF of 1 to the entire area. As a result, the BIAS of REF3 is very close to 0 due differing biases cancelling out each other."*

The quality of many of the figures needs to be improved. Figure 1 would benefit from a zoomed out map showing the relative locations of the study areas in the Swiss Alps and Qilian Mountains. In Figures 4, 10, and 11, are these means or medians shown with the red dots? Please include this information in the legends for these respective figures. In Figure 5 and in Figure 11, what time period is being shown for each of the stations? This information should to be included either in a separate table or in the figure captions. Finally, the latitude and longitude should be included directly on Figure 8, Figure 13, and Figure 2, rather than in the caption, in order to improve the figures' readability.

Please note that the number of plots are changed. The number used below are new ones.

Figure 1: A figure with the relative locations of the Swiss Alps and Qilian Mountains is added.

Figure 2: The elevation distribution from previous Figure 1 and a new plot of station used in different years are set as Figure 2.

Figure 5, 11 and 12: the red dots are median values, and this information will be added in the respective captions in a revised submission.

Figure 6 and 13: the observation periods are provided in Table 3.

Figure 9, 13 and A2: latitude and longitude are added.

[Figure]

Figure 1 Location of experimental region (a), observation stations in the Swiss Alps (b) and the Qilian Mountains (c).

[Figure]

Figure 2 Elevation distribution of observation stations (a), number of observation stations (N) used in different years (b).

[Figure]

Figure 9 Spatial variation of elevation (a), hypsometric position (b), normalized MRVBF (c) and *LSCF* (d) in selected slope terrain.

[Figure]

Figure 13 Fine-scale of land-surface influence ($ΔT^f$) for the test area.

[Figure]

Figure A2 Original MRVBF in the test area, Swiss Alps and the Qilian Mountains by using slope threshold of 50% and 16%.

Table 3. Comparison of observations against reference methods of REF2 and REF3.

| Station | Location | | | | Observation period | REF2 | | REF3 | |
|---------|----------|----------|----------|------------|--------------------|------|------|------|------|
|         | Lat (°)  | Lon (°)  | Ele (m)  | Topography |                    | RMSE | BIAS | RMSE | BIAS |
| COV     | 46.4180  | 9.8212   | 3351     | Peak       | 01/1998–12/2015    | 1.01 | 0.24 | 1.63 | -0.41 |
| DDS     | 38.0142  | 100.2421 | 4147     | Peak       | 10/2007–10/2009    | 1.34 | 0.56 | 2.38 | -1.05 |
| BEV1    | 46.5487  | 9.8538   | 2490     | Peak       | 09/1997–12/2015    | 1.22 | 0.54 | 1.41 | 0.04 |
| EBO     | 37.9492  | 100.9151 | 3294     | Slope      | 06/2013–12/2014    | 3.31 | 2.41 | 1.87 | 0.43 |
| SIA     | 46.4323  | 9.7623   | 1853     | Valley     | 01/1980–12/2015    | 2.44 | 1.14 | 1.65 | 0.50 |
| SAM     | 46.5263  | 9.8789   | 1756     | Valley     | 01/1980–12/2015    | 3.85 | 1.95 | 2.81 | 1.39 |

**Specific comments:**
Page 1, Line 18: Change "oder" to order.

We corrected the typo.

Page 6, Line 7: Include citation for "degree of valleyness."

Should it be Line 11? We named "degree of valleyness" (*V*) and descried by the normalized multiresolution index of valley bottom flatness (MRVBF) (Gallant & Dowling 2003). In this case, we added the citation of Gallant & Dowling (2003) before Eq. 10 rather than here. We hope you agree.

Page 13: Many of the references are missing doi numbers. Please include these.

Thank you for pointing this. The doi numbers will be added in a revised submission.

Page 25: I am unsure what you mean by "a quality of points visual."

Sorry for the typo, it should be *"a quantity of points visual"*.

References:

Daly, C., Gibson, W., Taylor, G., Johnson, G. & Pasteris, P. 2002. A knowledge-based approach to the statistical mapping of climate. *Climate Research* 22 : 99–113. DOI: 10.3354/cr022099

Daly, C., Taylor, G.H., Gibson, W.P., Parzybok, T.W., Johnson, G.L., Pasteris, P.A. & others 2000. High-quality spatial climate data sets for the United States and beyond. *Transactions of the ASAE-American Society of Agricultural Engineers* 43 : 1957–1962.

Gallant, J.C. & Dowling, T.I. 2003. A multiresolution index of valley bottom flatness for mapping depositional areas. *Water Resources Research* 39 : n/a–n/a. DOI: 10.1029/2002WR001426

Hijmans, R.J., Cameron, S.E., Parra, J.L., Jones, P.G. & Jarvis, A. 2005. Very high resolution interpolated climate surfaces for global land areas. *International journal of climatology* 25 : 1965–1978.

Thornton, P.E., Running, S.W. & White, M.A. 1997. Generating surfaces of daily meteorological variables over large regions of complex terrain. *Journal of Hydrology* 190 : 214–251. DOI: http://dx.doi.org/10.1016/S0022-1694(96)03128-9

---

## Referee Comment (RC2) · Anonymous Referee #2 · 20 Jun 2017

Air temperature downscaling is important for the mountain regions. This manuscript described a down scaling tools(maybe a software), which were validated by ground observed data in Apls and Qilian mountain. I have serveral concerns: (1) What is new? REDCAPP is a new one, or come from Fiddes and Gruber 2014; Gupta and Tarboton 2016, and Gao et. al. 2012? You should make a declaration clearly. (2) Data introduction is weak. specially, the description of DEM process is too short. I can not understand what you did on the DEM data. (3)Discussion is weak. (4) In general, REDCAPP can not be transfered to different regions. It is a big shortage of this method. So, you should give more work on this issue. otherwise, others can not use your verion 1.0.

---

## Author Comment (AC2) · 26 Jun 2017

**Response to Anonymous Referee #2**

Air temperature downscaling is important for the mountain regions. This manuscript described a down scaling tools (maybe a software), which were validated by ground observed data in Alps and Qilian mountain.

The authors would like to thank the reviewer for the constructive feedback, and the thorough assessment of the manuscript. Below we provide a point-to-point response to each comment, reviewer comments are given in black, responses are given in blue. Additionally, we have included details of how we intend to address these changes in a revised submission.

I have several concerns:

(1) What is new? REDCAPP is a new one, or come from Fiddes and Gruber 2014; Gupta and Tarboton 2016, and Gao et. al. 2012? You should make a declaration clearly.

Though the upper-air temperatures ($T_{pl}^{f}$ and $T_{pl}^{c}$) are obtained by following Fiddes & Gruber (2014) (as we described in first part of Section 4.1), REDCAPP is a new method extending previous work. This is because REDCAPP disaggregates the difference of upper-air and near-surface temperatures ($\Delta T$) as a proxy of surface effects. In the reference methods, surface effects are either ignored (REF2) or treated as spatially invariant at the fine scale (REF1 and REF3). But in REDCAPP, a fine-scale DEM-based land surface correction factor ($LSCF$) is simulated to derive the fine-scale surface effects caused by cold air pooling and topography influences (e.g. hypsometric position described in Section 2.2). In the revised manuscript, we will make a clear declaration of new issues of REDCAPP by comparing with the methods used by Fiddes & Gruber (2014), Gao, Bernhardt, & Schulz (2012) and Sen Gupta & Tarboton (2016) as the first paragraph of new Section of 6.1 "Comparison with other downscaling techniques". Furthermore, the manuscript title "Parameterizing valley inversions in air temperature data downscaled from re-analyses" points to the key difference with respect to earlier work.

*"Though the upper-air temperatures ($T_{pl}^{f}$ and $T_{pl}^{c}$) are obtained following Fiddes & Gruber (2014), disaggregating the difference of upper-air and near-surface temperatures as a proxy of surface effects ($\Delta T$) makes REDCAPP a new method. Additionally, the $\Delta T$ in REDCAPP is adjusted to fine scale responding to spatially heterogeneity of surface effect based on LSCF derived from DEM and observations, rather than ignored (REF2) or treated as spatial invariant (REF1 and REF3)."*

(2) Data introduction is weak. specially, the description of DEM process is too short. I can not understand what you did on the DEM data.

(a) We reformulated the Section of "Observations and quality control" (see below) to give a more detailed introduction of observations (Please also see our response to Referee #1).

*"The temperature from MeteoSwiss is observed using the Thygan instrument which has an accuracy of ± 0.01 °C, and temperatures from IMIS are measured by several different sensors (including Rotronic MP100H, Rotronic MP102H/HC2, Rotronic MP103A, Campbell Scientific CS215), with sensor accuracies ranging from ± 0.1 to ± 0.9 °C. In the Qilian Mountains, temperature sensor HMP155 with a typical accuracy of ± 0.2 °C are used. The 395 stations used cover an elevation range of ~250–4150 m as well as different topographic positions including peaks, slopes, plains and deep valleys (Figure 2a).*

*All temperature observations were filtered using a threshold (plausible values from -60 to 60 °C), and the outliers of temperature time series were removed by visually check. Time offsets between observations and ERA-Interim are avoided by conducting all analyses in UTC time. When using mean daily temperature, days with missing data were removed before further analysis. Though there are in total 395 stations used here, not all of them are available in a single year (Figure 2b). In total, there are ~2.5 × 10$^6$ observations of mean daily temperature in or after 1980 used here."*

(b) About the DEM noise I simply copied part of the ASTER Global Digital Elevation Model Version 2–Summary of Validation Results (Meyer et al., 2011) here

"the addition of higher-frequency topographic signal in GDEM2 as compared to GDEM1 came at the cost of added, nearly ubiquitous, high frequency noise, as is visually apparent and as indicated by the higher standard deviation of differences from benchmark elevations (USGS) and from SRTM postings (NGA) despite the general reduction of artifacts such as pits and spikes."

In this case, we aggregated the original ASTER DEM with a spatial resolution of 1 arc-second to a spacing of 3 arc-second to avoid the noise. We added a new figure in appendix as Figure A1 (see below) to support the introduction of DEM aggregation.

| 3270 | 3014 | 3465 | 3386 | 2810 | 4237 | 2210 | 2796 | 3147 |
| 2066 | 2783 | 2356 | 2699 | 3979 | 2556 | 4431 | 3736 | 3873 |
| 2844 | 3166 | 3864 | 4011 | 2517 | 3009 | 3421 | 4048 | 4206 |
| 2927 | 2104 | 2594 | 4016 | 2010 | 4054 | 3948 | 2345 | 3519 |
| 3532 | 3936 | 3555 | 2093 | 2478 | 2684 | 2877 | 2245 | 2837 |
| 2428 | 4095 | 2689 | 2655 | 4098 | 3374 | 3670 | 4224 | 2286 |
| 3894 | 4500 | 2923 | 2203 | 2782 | 2205 | 2846 | 2350 | 2912 |
| 2614 | 4451 | 2181 | 2468 | 4244 | 2954 | 4188 | 2172 | 4002 |
| 2026 | 2221 | 4449 | 4211 | 3203 | 4251 | 3716 | 3187 | 4446 |

| 2980 | 3244 | 3540 |
| 3095 | 3051 | 3105 |
| 3251 | 3169 | 3313 |

DEM of 1 arc-second                    DEM of 3 arc-second

Figure A1 Schematic illustration of DEM aggregation from a grid spacing of 1 arc-second to 3 arc-seconds by averaging. Numbers in the pixels are elevations in meter. In hypsometric simulation, the DEM with a grid spacing of 15 arc-second is derived using the same method.

(3) Discussion is weak.

(a) Section 6.1 Comparison with other downscaling techniques

We highlighted the new things of REDCAPP by comparing the reference methods used in this study in a new Section 6.1 "Comparison with the other downscaling techniques" (see below). Please also see the responses to comment (1). As mentioned by the Referee #1, a detailed discussion of comparisons of REDCAPP and exiting other methods, such as the Parameter-elevation Regression on Independent Slopes Model (PRISM) (Daly et al., 2000; Daly et al., 2002), the Daily Surface Weather and Climatological Summary (DAYMET) (e.g., Thornton et al., 1997), and the techniques used in Hijmans et al. (2005) is needed to highlight the advantages of REDCAPP. In this case, we added a detailed discussion (see below) in the Section 6.1.

6.1 Comparison with other downscaling techniques

*Though the upper-air temperatures ($T_{pl}^{f}$ and $T_{pl}^{c}$) is obtained by following Fiddes & Gruber (2014), disaggregating the difference of upper-air and near-surface temperatures as a proxy of surface effects*

*(ΔT) makes REDCAPP a new method. Additionally, the ΔT in REDCAPP is adjusted to fine scale responding to spatially heterogeneity of surface effect based on LSCF derived from DEM, rather than ignored (REF2) or treated as spatial invariant (REF1 and REF3).*

*Besides the lapse rate correction methods referenced in this study, many existing downscaling approaches for mountainous terrain focus on deriving fine-scale T through interpolation (e.g. truncated Gaussian weighting filter, Inverse Distance Weighting, or Kriging) of surrounding observations, and adjustments are then made based on fine-scale topography. PRISM (Parameter-elevation Regressions on Independent Slopes Model) (Daly et al., 2000; Daly, Gibson, Taylor, Johnson, & Pasteris, 2002), for example, derives a weighing function to represent the relationship of T with geographic (e.g. slopes, coastal) and meteorological (e.g. atmosphere boundary-layer) factors. Similarly, the approach by Thornton et al. (1997) calculates interpolation weights for the stations nearby, and corrected the downscaled results based on an empirical relationship of T to elevation, and Hijmans et al. (2005) conducted a second-order spline interpolation using latitude, longitude and elevation as independent variables. As observations are usually sparse in mountains, especially at higher elevation, these methods are expected to often have significant uncertainty caused by inadequately sampling of elevation and hence lapse rate. In comparison, REDCAPP relies on reanalysis data for air temperature and uses station data only for calibration of the LSCF related to CAP. REDCAPP derives lapse rates from multiple layers of upper air temperature encompassing the entire elevation range of study area. Thus, REDCAPP results are expected to be robust because both the Tsa and Tpl from reanalysis are used.*

(b) Section 6.3 Transferability

The discussion of model transferability is reinforced by clarifying the differences of transferring parameters of *LSCF* and applying REDCAPP in other mountains. Details please see our response to comment (4).

(4) In general, REDCAPP can not be transferred to different regions. It is a big shortage of this method. So, you should give more work on this issue. otherwise, others can not use your version 1.0.

In section 6.3 Transferability, we give a discussion of transferability of parameter values (α, β, γ in Eq. 6 and 7) of establishing land surface correction factor (LSCF). The parameters of land surface correction factor (LSCF) is different between Alps and Qilian Mountains, and hence hard to be directly transferred from the tested two mountains to other regions. But for REDCAPP, it could be applied in other mountains. This is because the establishment of LSCF is derived from fine-scale

DEM and observations, this fundamental concept is physically sensible and could be used in different mountains.

We reformulated the sentence of

*"The difference in estimated parameter values limits the transferability of REDCAPP as it requires tests new mountain regions to investigate suitability."*

to

*"The difference in estimated parameter values of LSCF limits the directly transferability of REDCAPP parameters from the mountain regions tested here to others as it requires new calibration in other mountain regions."* to clarify

Additionally, we added

*"REDCAPP can be applied to other mountains once the parameters (α, β and γ in Eq. 6 and 7) of LSCF are derived based on observations and a fine-scale DEM."*

at the end of Section 6.3 Transferability to clarify the difference of parameter transferability and applying models in other regions.

---

## Author Response (AR1)

**Author's Response on "REDCAPP (v1.0): Parameterizing valley inversions in air temperature data downscaled from re-analyses"**

Bin Cao[1,2], Stephan Gruber[2], and Tingjun Zhang[1]

[1]Key Laboratory of Western China's Environmental Systems (MOE), College of Earth and Environmental Sciences, Lanzhou University, Lanzhou 730000, China
[2]Department of Geography & Environmental Studies, Carleton University, Ottawa, K1S 5B6, Canada

*Correspondence to:* Bin Cao (caob08@lzu.edu.cn)

The authors would like to thank Dr. Patrick Jöckel and two anonymous referees for the constructive feedback, and the thorough assessment of the manuscript. Below we provide a point-to-point response to each comment, reviewer comments are given in black, responses are given in blue. Additionally, we have included details of how we address these changes in a revised submission.

**1   Response to Anonymous Referee #1**

**1.1   General comments**

I am missing a discussion of how the method that the authors developed in the present manuscript differs from other downscaling approaches that already exist for complex or mountainous terrain. There needs to be more discussion of how the

10   authors' method improves upon and is better than pre-existing downscaling techniques that also use variable lapse rates and incorporate information about the land surface and topographical characteristics, e.g. the Parameter-elevation Regression on Independent Slopes Model (PRISM) (Daly et al., 2000; Daly et al., 2002), the Daily Surface Weather and Climatological Summary (DAYMET) (e.g., Thornton et al., 1997), and the techniques used in Hijmans et al. (2005).

15   We added a detailed discussion (see below) of comparisons of REDCAPP and existing methods (e.g. PRISM, DAYMET and other approaches) as a new Section 6.1 "Comparison with other downscaling techniques."

*6.1 Comparison with other downscaling techniques*

*Besides the lapse rate correction methods referenced in this study, many existing downscaling approaches for mountainous ter-*

20   *rain focus on deriving fine-scale $T$ through interpolation (e.g. truncated Gaussian weighting filter, Inverse Distance Weighting, or Kriging) of surrounding observations, and adjustments are then made based on fine-scale topography. PRISM (Parameter-elevation Regressions on Independent Slopes Model) (Daly et al., 2000, 2002), for example, derives a weighing function to represent the relationship of $T$ with geographic (e.g. slopes, coastal) and meteorological (e.g. atmosphere boundary-layer)*

*factors . Similarly, the approach by Thornton et al. (1997) calculates interpolation weights for the stations nearby, and corrected the downscaled results based on an empirical relationship of $T$ to elevation, and Hijmans et al. (2005) conducted a second-order spline interpolation using latitude, longitude and elevation as independent variables. As observations are usually sparse in mountains, especially at higher elevation, these methods are expected to have significant uncertainty caused by inadequately sampling of elevation and hence lapse rate. In comparison, REDCAPP relies on reanalysis data for air temperature and uses station data only for calibration of the LSCF related to CAP. REDCAPP derives lapse rates from multi layers of upper air temperature encompassing the entire elevation range of study area. Thus, REDCAPP results are expected to be robust because both the $T_{sa}$ and $T_{pl}$ from reanalysis are used.*

The temperature lapse rate is defined as decreasing with height and thus a negative lapse rate implies a temperature increase with height. This change needs to be implemented throughout the manuscript.

Thank you for pointing this out. We corrected throughout the manuscript, changes are listed below:

1. *"For example, Lewkowicz and Bonnaventure (2011) reported that average lapse rates could be positive in mountains due to strong winter inversion and result in lower T in valleys than at higher locations."*
   is changed to
   *"For example, Lewkowicz and Bonnaventure (2011) reported that T in valleys was lower than at higher locations in mountains due to strong winter inversion."*

2. *"This is because the lapse rates are expected to decrease owing to the presence of CAPs."*
   is changed to
   *"This is because the lapse rates are expected to **increase** owing to the presence of CAPs."*

In Section 3.2, more discussion is required about the meteorological stations that the authors used, e.g., instrument type, completeness of the data sets at these stations within the two study regions, etc. Also, the authors mention in line 3 of page 5 that mean daily temperatures in 1980 or after are used. The total time period of the study needs to be indicated. I am also not sure what the authors mean by "obviously wrong values" on pages 4-5 in this section. More description is necessary here too.

Yes, we agree the discussion of observations will benefit from more detail. We reformulated this part by adding the instrument type and accuracy (see below). Additionally, a figure showing observation completeness is added in Figure 2.

The "obviously wrong values" means the value out of the range of -60 to 60 °C or not consistent by comparing values with the day before and after the checking day.

[Figure]

**Figure 2.** Elevation distribution of observation station (a), number of observation stations (N) used in different years (b).

*The temperature from MeteoSwiss is observed using the Thygan instrument which has an accuracy of ± 0.01 °C, and temperatures from IMIS are measured by several different sensors (including Rotronic MP100H, Rotronic MP102H/HC2, Rotronic MP103A, Campbell Scientific CS215), with sensor accuracies ranging from ± 0.1 to ± 0.9 °C. In the Qilian Mountains, temperature sensor HMP155 with a typical accuracy of ± 0.2 °C are used. The 395 stations used cover an elevation range of*

5  *250–4150 m as well as different topographic positions including peaks, slopes, plains and deep valleys (Figure 2a).*

 *All temperature observations were filtered using a threshold (plausible values from -60 to 60 °C), and the outliers of temperature time series were removed by visually check. Time offsets between observations and ERA-Interim are avoided by conducting all analyses in UTC time. When using mean daily temperature, days with missing data were removed before further analysis. Though there are in total 395 stations used here, not all of them are available in a single year (Figure 2b). In total, there are*

10  $2.5 \times 10^6$ *observations of mean daily temperature in or after 1980 used here.*

In Section 5.3.1, the authors note that the bias in the Swiss Alps increases with the implementation of the REDCAPP technique, but no explanation is offered as to why this is.

15  To clarify, we added

*This is because REF3 resulted in air temperatures being too low at high elevation, while the influence of CAP was underestimated in valleys by applying a fixed LSCF of 1 to the entire area. As a result, the BIAS of REF3 is very close to 0 due differing biases cancelling out each other.*

The quality of many of the figures needs to be improved. Figure 1 would benefit from a zoomed out map showing the relative locations of the study areas in the Swiss Alps and Qilian Mountains. In Figures 4, 10, and 11, are these means or medians shown with the red dots? Please include this information in the legends for these respective figures. In Figure 5 and in Figure 11, what time period is being shown for each of the stations? This information should to be included either in a separate table or in the figure captions. Finally, the latitude and longitude should be included directly on Figure 8, Figure 13, and Figure 2, rather than in the caption, in order to improve the figures' readability.

Please note that the number of plots are changed. The number used below are new ones.

1. Figure 1: A figure with the relative locations of the Swiss Alps and Qilian Mountains is added.

2. Figure 2: The elevation distribution from previous Figure 1 and a new plot of station used in different years are set as Figure 2.

3. Figure 5, 11 and 12: the red dots are median values, and this information is added in the respective captions in a revised submission.

4. Figure 6 and 13: the observation periods are provided in Table 3.

5. Figure 9, 13 and A2: latitude and longitude are added.

[Figure]

**Figure 1.** Location of experimental region (a), observation stations in the Swiss Alps (b) and the Qilian Mountains (c).

[Figure]

**Figure 9.** Spatial variation of elevation (a), hypsometric position (b), normalized MRVBF (c) and $LSCF$ (d) in selected slope terrain.

[Figure]

**Figure 10.** Fine-scale of land-surface influence ($\Delta T^f$) for the test area.

[Figure]

**Figure A3.** Original MRVBF in the test area, Swiss Alps and the Qilian Mountains by using slope threshold of 50% and 16%.

**Table 3.** Comparison of observations against reference methods of REF2 and REF3.

| Station | Location | | | | Observation period | REF2 | | REF3 | |
| --- | --- | --- | --- | --- | --- | --- | --- | --- | --- |
| | Lat (°) | Lon (°) | Ele (m) | Topography | | RMSE | BIAS | RMSE | BIAS |
| COV | 46.4180 | 9.8212 | 3351 | Peak | 01/1998–12/2015 | 1.01 | 0.24 | 1.63 | -0.41 |
| DDS | 38.0142 | 100.2421 | 4147 | Peak | 10/2007–10/2009 | 1.34 | 0.56 | 2.38 | -1.05 |
| BEV1 | 46.5487 | 9.8538 | 2490 | Peak | 09/1997–12/2015 | 1.22 | 0.54 | 1.41 | 0.04 |
| EBO | 37.9492 | 100.9151 | 3294 | Slope | 06/2013–12/2014 | 3.31 | 2.41 | 1.87 | 0.43 |
| SIA | 46.4323 | 9.7623 | 1853 | Valley | 01/1980–12/2015 | 2.44 | 1.14 | 1.65 | 0.50 |
| SAM | 46.5263 | 9.8789 | 1756 | Valley | 01/1980–12/2015 | 3.85 | 1.95 | 2.81 | 1.39 |

**1.2 Specific comments**

Page 1, Line 18: Change "oder" to order.

We corrected the typo.

5 Page 6, Line 7: Include citation for "degree of valleyness."

Should it be Line 11? We named "degree of valleyness" (V) and descried by the normalized multiresolution index of valley bottom flatness (MRVBF) (Gallant and Dowling, 2003). In this case, we added the citation of Gallant and Dowling (2003) before Eq. 10 rather than here. We hope you agree.

10 Page 13: Many of the references are missing doi numbers. Please include these.

Thank you for pointing this. The doi numbers are added in a revised submission.

Page 25: I am unsure what you mean by "a quality of points visual."

Sorry for the typo, it should be "a quantity of points visual".

**2 Response to Referee #2**

**2.1 General Comments**

(1) Air temperature downscaling is important for the mountain regions. This manuscript described a down scaling tools (maybe a software), which were validated by ground observed data in Alps and Qilian mountain.I have several concerns:

What is new? REDCAPP is a new one, or come from Fiddes and Gruber 2014; Gupta and Tarboton 2016, and Gao et. al. 2012? You should make a declaration clearly.

Though the upper-air temperatures ($T_{pl}^f$ and $T_{pl}^c$) are obtained by following Fiddes and Gruber (2014) (as we described in first part of Section 4.1), REDCAPP is a new method extending previous work. This is because REDCAPP disaggregates the difference of upper-air and near-surface temperatures ($\Delta T$) as a proxy of surface effects. In the reference methods, surface effects are either ignored (REF2) or treated as spatially invariant at the fine scale (REF1 and REF3). But in REDCAPP, a fine-scale DEM-based land surface correction factor (LSCF) is simulated to derive the fine-scale surface effects caused by cold air pooling and topography influences (e.g. hypsometric position described in Section 2.2). In the revised manuscript, we made a clear declaration of new issues of REDCAPP by comparing with the methods used by Fiddes and Gruber (2014); Gao et al. (2012); Gupta and Tarboton (2016), as the first paragraph of new Section of 6.1 "Comparison with other downscaling tech- niques" (see below). Furthermore, the manuscript title "Parameterizing valley inversions in air temperature data downscaled from re-analyses" points to the key difference with respect to earlier work.

*Though the upper-air temperatures ($T_{pl}^f$ and $T_{pl}^c$) are obtained following Fiddes and Gruber (2014), disaggregating the dif- ference of upper-air and near-surface temperatures as a proxy of surface effects ($\Delta T$) makes REDCAPP a new method. Additionally, the $\Delta T$ in REDCAPP is adjusted to fine scale responding to spatially heterogeneity of surface effect based on LSCF derived from DEM and observations, rather than ignored (REF2) or treated as spatial invariant (REF1 and REF3).*

(2) Data introduction is weak. specially, the description of DEM process is too short. I can not understand what you did on the DEM data.

1. We reformulated the Section of "Observations and quality control" (see below) to give a more detailed introduction of observations (Please also see our response to Referee #1)

*The temperature from MeteoSwiss is observed using the Thygan instrument which has an accuracy of ± 0.01 °C, and temperatures from IMIS are measured by several different sensors (including Rotronic MP100H, Rotronic MP102H/HC2, Rotronic MP103A, Campbell Scientific CS215), with sensor accuracies ranging from ± 0.1 to ± 0.9 °C. In the Qilian Mountains, temperature sensor HMP155 with a typical accuracy of ± 0.2 °C are used. The 395 stations used cover an elevation range of 250–4150 m as well as different topographic positions including peaks, slopes, plains and deep valleys (Figure 2a).*
*All temperature observations were filtered using a threshold (plausible values from -60 to 60 °C), and the outliers of temperature time series were removed by visually check. Time offsets between observations and ERA-Interim are avoided by conducting all analyses in UTC time. When using mean daily temperature, days with missing data were*

| 3270 | 3014 | 3465 | 3386 | 2810 | 4237 | 2210 | 2796 | 3147 |
| 2066 | 2783 | 2356 | 2699 | 3979 | 2556 | 4431 | 3736 | 3873 |
| 2844 | 3166 | 3864 | 4011 | 2517 | 3009 | 3421 | 4048 | 4206 |
| 2927 | 2104 | 2594 | 4016 | 2010 | 4054 | 3948 | 2345 | 3519 |
| 3532 | 3936 | 3555 | 2093 | 2478 | 2684 | 2877 | 2245 | 2837 |
| 2428 | 4095 | 2689 | 2655 | 4098 | 3374 | 3670 | 4224 | 2286 |
| 3894 | 4500 | 2923 | 2203 | 2782 | 2205 | 2846 | 2350 | 2912 |
| 2614 | 4451 | 2181 | 2468 | 4244 | 2954 | 4188 | 2172 | 4002 |
| 2026 | 2221 | 4449 | 4211 | 3203 | 4251 | 3716 | 3187 | 4446 |

DEM of 1 arc-second

| 2980 | 3244 | 3540 |
| 3095 | 3051 | 3105 |
| 3251 | 3169 | 3313 |

DEM of 3 arc-second

**Figure A1.** Schematic illustration of DEM aggregation from a grid spacing of 1 arc-second to 3 arc-second by averaging. Numbers in the pixels are elevations in meter. In hypsometric simulation, the DEM with a grid spacing of 15 arc-second is derived using the same method.

*removed before further analysis. Though there are in total 395 stations used here, not all of them are available in a single year (Figure 2b). In total, there are $2.5 \times 10^6$ observations of mean daily temperature in or after 1980 used here.*

5   2. About the DEM noise I simply copied part of the ASTER Global Digital Elevation Model Version 2–Summary of Validation Results (Meyer et al., 2011) here

*"the addition of higher-frequency topographic signal in GDEM2 as compared to GDEM1 came at the cost of added, nearly ubiquitous, high frequency noise, as is visually apparent and as indicated by the higher standard deviation of*
10   *differences from benchmark elevations (USGS) and from SRTM postings (NGA) despite the general reduction of artifacts such as pits and spikes."*

In this case, we aggregated the original ASTER DEM with a spatial resolution of 1 arc-second by averaging to a spacing of 3 arc-second to avoid the noise. We added a new figure in appendix as Figure A1 to support the introduction of DEM
15   aggregation.

(3) Discussion is weak.

1. Section 6.1 Comparison with other downscaling techniques

   We highlighted the new things of REDCAPP by comparing the reference methods used in this study in a new Section 6.1 "Comparison with the other downscaling techniques" (see below). Please also see the responses to comment (1). As mentioned by the Referee #1, a detailed discussion of comparisons of REDCAPP and exiting other methods, such as the Parameter-elevation Regression on Independent Slopes Model (PRISM) (Daly et al., 2000, 2002), the Daily Surface Weather and Climatological Summary (DAYMET) Thornton et al. (1997), and the techniques used in Hijmans et al. (2005) is needed to highlight the advantages of REDCAPP. In this case, we added a detailed discussion (see below) in the Section 6.1.

   *6.1 Comparison with other downscaling techniques*

   *Though the upper-air temperatures ($T_{pl}^{f}$ and $T_{pl}^{c}$) are obtained following Fiddes and Gruber (2014), disaggregating the difference of upper-air and near-surface temperatures as a proxy of surface effects ($\Delta T$) makes REDCAPP a new method. Additionally, the $\Delta T$ in REDCAPP is adjusted to fine scale responding to spatially heterogeneity of surface effect based on LSCF derived from DEM and observations, rather than ignored (REF2) or treated as spatial invariant (REF1 and REF3).*

   *Besides the lapse rate correction methods referenced in this study, many existing downscaling approaches for mountainous terrain focus on deriving fine-scale $T$ through interpolation (e.g. truncated Gaussian weighting filter, Inverse Distance Weighting, or Kriging) of surrounding observations, and adjustments are then made based on fine-scale topography. PRISM (Parameter-elevation Regressions on Independent Slopes Model) (Daly et al., 2000, 2002), for example, derives a weighing function to represent the relationship of $T$ with geographic (e.g. slopes, coastal) and meteorological (e.g. atmosphere boundary-layer) factors . Similarly, the approach by Thornton et al. (1997) calculates interpolation weights for the stations nearby, and corrected the downscaled results based on an empirical relationship of $T$ to elevation, and Hijmans et al. (2005) conducted a second-order spline interpolation using latitude, longitude and elevation as independent variables. As observations are usually sparse in mountains, especially at higher elevation, these methods are expected to have significant uncertainty caused by inadequately sampling of elevation and hence lapse rate. In comparison, REDCAPP relies on reanalysis data for air temperature and uses station data only for calibration of the LSCF related to CAP. REDCAPP derives lapse rates from multi layers of upper air temperature encompassing the entire elevation range of study area. Thus, REDCAPP results are expected to be robust because both the $T_{sa}$ and $T_{pl}$ from reanalysis are used.*

2. Section 6.3 Transferability

   The discussion of model transferability is reinforced by clarifying the differences of transferring parameters of $LSCF$ and applying REDCAPP in other mountains. Details please see our response to comment (4).

(4) In general, REDCAPP can not be transferred to different regions. It is a big shortage of this method. So, you should give more work on this issue. otherwise, others can not use your version 1.0.

In section 6.3 Transferability, we give a discussion of transferability of parameter values ($\alpha$, $\beta$ and $\gamma$ in Eq. 6 and 7) of establishing land surface correction factor ($LSCF$). The parameters of land surface correction factor ($LSCF$) is different between Alps and Qilian Mountains, and hence hard to be directly transferred from the tested two mountains to other regions. But for REDCAPP, it could be applied in other mountains. This is because the establishment of $LSCF$ is derived from fine-scale DEM and observations, this fundamental concept is physically sensible and could be used in different mountains.

We reformulated the sentence of

*"The difference in estimated parameter values limits the transferability of REDCAPP as it requires tests new mountain regions to investigate suitability."*
to

*"The difference in estimated parameter values of LSCF limits the directly transferability of REDCAPP parameters from the mountain regions tested here to others as it requires new calibration in other mountain regions."* to clarify.

Additionally, we added

*"REDCAPP can be applied to other mountains once the parameters ($\alpha$, $\beta$ and $\gamma$ in Eq. 6 and 7) of LSCF are derived based on observations and a fine-scale DEM."*

at the end of Section 6.3 Transferability to clarify the difference of parameter transferability and applying models in other regions.

**References**

[revised manuscript text omitted]